# DiffPC: Diffusion-based High Perceptual Fidelity Image Compression with Semantic Refinement

**Yichong Xia**[1,3,†], **Yimin Zhou**[1,†], **Jinpeng Wang**[1], **Baoyi An**[4], **Haoqian Wang**[1], **Bin Chen**[2,3*]

[1]Tsinghua Shenzhen International Graduate School, [2]Harbin Institute of Technology, Shenzhen
[3]Peng Cheng Laboratory, [4]Huawei Technologies Company Ltd.

## ABSTRACT

Reconstructing high-quality images under low bitrates conditions presents a challenge, and previous methods have made this task feasible by leveraging the priors of diffusion models. However, the effective exploration of pre-trained latent diffusion models and semantic information integration in image compression tasks still needs further study. To address this issue, we introduce **Diff**usion-based High **P**erceptual Fidelity Image **C**ompression with Semantic Refinement (DiffPC), a two-stage image compression framework based on stable diffusion. DiffPC efficiently encodes low-level image information, enabling the highly realistic reconstruction of the original image by leveraging high-level semantic features and the prior knowledge inherent in diffusion models. Specifically, DiffPC utilizes a multi-feature compressor to represent crucial low-level information with minimal bitrates and employs pre-embedding to acquire more robust hybrid semantics, thereby providing additional context for the decoding end. Furthermore, we have devised a control module tailored for image compression tasks, ensuring structural and textural consistency in reconstruction even at low bitrates and preventing decoding collapses induced by condition leakage. Extensive experiments demonstrate that our method achieves state-of-the-art perceptual fidelity and surpasses previous perceptual image compression methods by a significant margin in statistical fidelity.

## 1 INTRODUCTION

In this epoch of rapid multimedia advancement, the constraints of limited network bandwidth and costly storage hinder the transmission and preservation of large-scale high-definition raw images, rendering image compression algorithms with high compression rates increasingly crucial. Traditional compression standards (Bellard; Wallace, 1992) employ manually crafted transformations to seek compression representations, yet these methods may exhibit severe block artifacts or even chromatic aberrations at low bitrates ($\leq 0.2$ bpp). Neural image compression algorithms (Ballé et al., 2016; Cheng et al., 2020; He et al., 2022), based on end-to-end optimization schemes for rate-distortion have demonstrated superior performance compared to traditional compression standards. Nonetheless, these distortion-driven approaches may still yield displeasing blurriness in scenarios with constrained bandwidth. At low bitrates, the relevance of pixel-level metrics such as Mean Squared error (MSE) as an evaluation criterion drops, directly resulting in the loss of substantial texture details and realism (also known as perceptual fidelity)[1] in such compression schemes. (Blau & Michaeli, 2019; Agustsson et al., 2023) characterize this phenomenon as a triple trade-off between bit rate, distortion, and realism. This signifies that seeking compact representations at low bit rates in the pixel domain inevitably leads to a decline in human perception and a lack of human observer's image semantic consistency. These semantic deficiencies diminish the practicality of compression algorithms at low bitrates, thus making exploring the trade-offs between bit rate and perceptual fidelity an imperative subject of study.

---

[*]Corresponding author(s). [†] Equal contribution.
[1]We will utilize this pair of synonyms simultaneously, both of which can be measured using perceptual metrics such as LPIPS and FID.

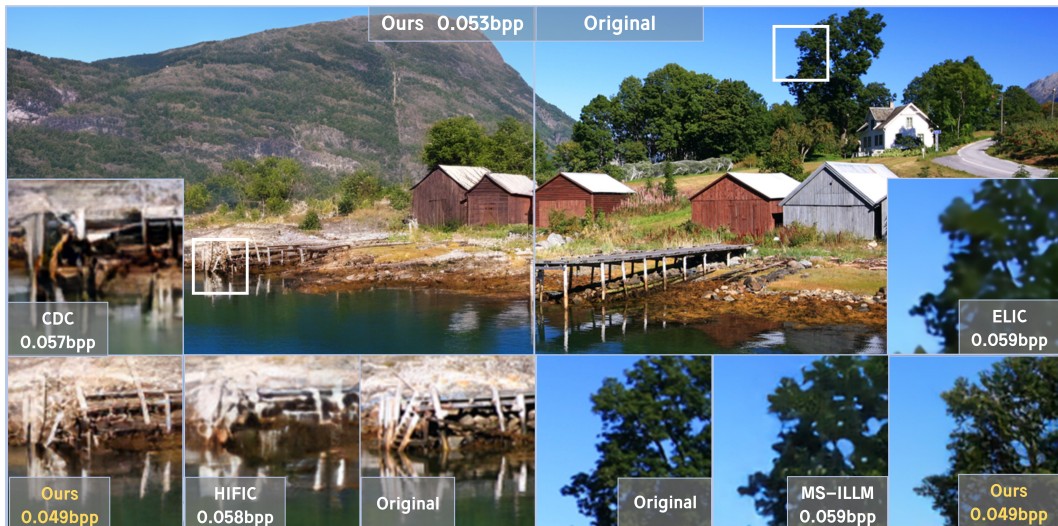

Figure 1: Qualitative comparison between HiFiC (Mentzer et al., 2020), MS-ILLM (Muckley et al., 2023) , ELIC (He et al., 2022), CDC (Yang & Mandt, 2024), and our proposed approach. DiffPC (Ours) is capable of reconstructing complex textures with realism, even at extremely low bit rates. In contrast, both HiFiC and MS-ILLM exhibit significant artifacts and texture loss. *Best viewed when zoomed in.*

The encoder-decoder architecture enhances the realism of decoded images by introducing perceptual loss and generative adversarial frameworks (Mentzer et al., 2020; Muckley et al., 2023). These efforts partially compensate for the semantic information and texture lost in distortion-driven schemes. Yet, excessive reliance on handcrafted losses leads to inevitable artifacts in low bitrate scenarios, diminishing the model's statistical fidelity. Compared to GAN, diffusion models have proven to generate high-realism images with enhanced statistical fidelity over the years (Dhariwal & Nichol, 2021; Rombach et al., 2022). This suggests that diffusion models are better suited for low bitrate image compression scenarios. (Yang & Mandt, 2024) realizes this concept by employing the DDPM framework (Ho et al., 2020) in the image domain and achieves superior result. However, these endeavors entail substantial time investment for retraining diffusion components, and the model's generalization is highly constrained by training data and computational resources.

Recently, the text-to-image Latent Diffusion Model (LDM) (Rombach et al., 2022) has further explored the potential of diffusion models in generative tasks. These data-driven foundational diffusion models have proven to offer strong priors for various visual tasks such as image segmentation (Tian et al., 2024) and image super-resolution (Lin et al., 2023) and possess the capability to fuse multimodal semantic information. (Lei et al., 2023) have attempted to control pre-trained LDMs by encoding sketches and textual semantics, sampling image recovery during the decoding process within the diffusion framework. Given the challenges in jointly optimizing semantic embeddings and compressing representations of target transmissions, they have invested significant resources in iterative semantic embedding and alignment. Furthermore, while encoding solely semantic information significantly reduces the bitrate, the trade-off dramatically decreases fidelity. (Careil et al., 2024) have employed fully trained conditional LDM and encoded conditional images using trainable codebooks. On one hand, this requires meticulous model optimization on datasets consisting of millions of images; on the other hand, VQ-based compressors that exclusively accept latent inputs cannot recover information lost during the encoding process in the LDM encoder, resulting in suboptimal bitrate allocation during compression.

To address the challenges above, we have proposed the **Diff**usion-based High **P**erceptual Fidelity Image **C**ompression with Semantic Refinement (DiffPC). DiffPC leverages image-level control flow and semantic-level control flow to faithfully reconstruct highly realistic decoded images. At the encoding end, we employ a multi-feature compressor that can reconstruct high-quality image-level control flows even at extremely low bit rates. Furthermore, we have devised IC-ControlNet to enable pixel-level precision control for these low-level image controls. At the decoding end, to compensate

for semantic deficiencies leading to edge distortions and texture losses, we have devised a pre-embedding module to efficiently generate robust hybrid semantics: textual semantic signals of the images are pre-modulated with low-level image signals and injected into the diffusion model through a linear mapping layer. With these characteristics, DiffPC can achieve superior perceptual and statistical fidelity even at low bit rates, effectively recovering texture details that align more closely with human perception, as illustrated in Figure 1.

In sum, our contributions are as follows:

- We propose a two-stage lossy image compression framework. In the first stage, DiffPC can attain superior bitrate allocation and reconstruct a more precise *low-level image control branch* guided by importance-weighted losses. In the second stage, DiffPC efficiently integrates a *high-level semantic control branch* to exert more precise control.

- We devise IC-ControlNet (ICCN) to utilize a low-level image control branch and effectively integrate the robust prior of pre-trained LDM. ICCN enables precise control over the sampling process, ensuring the denoising model faithfully recovers the pre-compressed image.

- We introduce a hybrid semantic refinement module to generate a high-level semantic control branch. This module adopts a pre-embedding approach that combines textual semantics with visual semantics, producing a hybrid semantic representation that is easily injectable into the denoising model.

## 2 RELATED WORK

**Learned lossy Image Compression** The advancement of deep neural networks has spurred the emergence of deep compression algorithms, showcasing performance surpassing traditional image compression standards like JPEG (Wallace, 1992). (Ballé et al., 2016)pioneered an end-to-end model based on autoencoders. (Ballé et al., 2018) introduced a hyperprior entropy model, while (Minnen et al., 2018) elevated performance significantly by introducing prior through autoregressive context modeling at the expense of decoding complexity. (Cheng et al., 2020) employed a discrete mixture model to more precisely model the latent distribution. (He et al., 2022), based on the orthogonality of features in channel and spatial dimensions, devised an asymmetric autoregressive entropy encoder. Based on Shannon's rate-distortion theory (Shannon, 1948), the optimization objectives of these approaches can be formalized as follows:

$$\mathcal{L} = R(\hat{\boldsymbol{y}}) + \lambda D\left(\boldsymbol{x}, \hat{\boldsymbol{x}}\right). \tag{1}$$

In the above, $\hat{\boldsymbol{y}}$ represents the distorted latent representation of the input image $\boldsymbol{x}$, and $R(\hat{\boldsymbol{y}})$ estimates the entropy of $\hat{\boldsymbol{y}}$ to provide the required number of bits for encoding. $D\left(\boldsymbol{x}, \hat{\boldsymbol{x}}\right)$ measures the distortion of the reconstructed image, which is often defined as pixel-wise loss (i.e., MSE). As (Blau & Michaeli, 2019) has indicated that solely optimizing the MSE-bitrate function can severely compromise the statistical fidelity of compressed data, (Mentzer et al., 2020) proposed the use of generative adversarial network structures combined with adversarial and perceptual losses, effectively enhancing the realism of decoded images. Subsequently, (Muckley et al., 2023) introduced a discriminator based on local image representations, greatly improving statistical fidelity. Subsequent work has explored image compression at ultra-low bitrates within generative architectures (Jiang et al., 2023; Mao et al., 2024; Lu et al., 2024). (Lee et al., 2024) enhances perceptual fidelity by incorporating additional textual semantics into the GAN structure. Influenced by the development of diffusion models, (Hoogeboom et al., 2023) adopted the ddpm architecture, using a neural autoencoder as a baseline compressor. Building upon this, (Yang & Mandt, 2024; Kuang et al., 2024) replaced the compressor with a VAE structure with a hyperprior and jointly trained a denoising network and compressor, none of them consider the use of additional bits for transmitting semantic information. (Lei et al., 2023) utilized a text-to-image latent diffusion model, engaging in time-consuming image captioning through prompt inversion iterations and adding image sketch assistance in decoding, enabling image compression at extremely low bit rates. (Careil et al., 2024) employed a fully trained LDM, achieving better performance gains at a high training cost. (Li et al., 2024b) leveraged a pre-trained diffusion model prior but overlooked high-level semantics and bitrate allocation at the encoding end, resulting in suboptimal performance.

**Diffusion Model** The diffusion model utilizes priors from non-equilibrium statistical physics to transform the data distribution $z_0$ to a known distribution $z_T$ (typically a Gaussian distribution) through a Markov chain. The forward process of this Markov chain $q(z_t \mid z_{t-1})$ is defined as gradually adding manually designed noise and then fitting the reverse sampling process $p(z_{t-1} \mid z_t)$ through a neural network $\mathcal{M}_\theta(\cdot)$. Specifically, the forward and reverse processes are defined as:

$$q(z_t \mid z_{t-1}) \quad = \mathcal{N}\left(z_t \mid \sqrt{1-\beta_t}z_{t-1}, \beta_t \mathbf{I}\right), \tag{2}$$

$$p_\theta(z_{t-1} \mid z_t) \quad = \mathcal{N}\left(z_{t-1} \mid \mathcal{M}_\theta(z_t, t), \Sigma_\theta(z_t, t)\right). \tag{3}$$

(Ho et al., 2020) improved upon the original diffusion probabilistic model and demonstrated that optimizing the Evidence Lower Bound (ELBO) of the data distribution is equivalent to optimizing the following objective:

$$\mathcal{L}_{\text{Diff}} = \mathbb{E}_{z_0, t, \epsilon} \left\| \epsilon - \mathcal{DN}_\theta\left(\sqrt{\bar{\alpha}_t}z_0 + \sqrt{1-\bar{\alpha}_t}\epsilon, t\right) \right\|^2, \tag{4}$$

where $\mathcal{DN}_\theta(\cdot)$ stands for denoising network. (Ho & Salimans, 2022) devised a framework for conditional diffusion models, outperforming GANs in image generation. By employing large-scale text encoder networks, conditional diffusion models can generate high-realism images from natural language prompts (Rombach et al., 2022). The Latent Diffusion Model (LDM) utilizes autoencoders to confine the denoising process within the low-dimensional embeddings of the data, significantly reducing complexity. Building upon the LDM framework, Stable Diffusion (Rombach et al., 2022) established a large-scale text-to-image latent diffusion model capable of receiving multimodal control inputs and consistently producing high-fidelity images. Stable Diffusion (SD) employs a pre-trained encoder $\mathcal{E}$ to encode an image $x$ into a latent variable $z_0 = \mathcal{E}(x)$. Subsequently, SD performs a noise addition and denoising process similar to 3 in the latent space. Similar to 4, the optimization objective for Stable Diffusion is:

$$\mathcal{L}_{sd} = \mathbb{E}_{z_0, c, s, t, \epsilon} \left[ \left\| \epsilon - \mathcal{DN}_\theta\left(\sqrt{\bar{\alpha}_t}z_0 + \sqrt{1-\bar{\alpha}_t}\epsilon, c, s, t\right) \right\|_2^2 \right]. \tag{5}$$

Here, $c$ and $s$ respectively refer to low-level image controls (e.g., image contours, image degradation), and high-level semantic controls (e.g., textual descriptions of images, category label). Our work is built upon the framework of Stable Diffusion, leveraging the robust priors of large-scale LDM on natural images and their semantic fusion capabilities.

## 3 METHODOLOGY

### 3.1 OVERALL FRAMEWORK

Our DiffPC framework is illustrated as Figure 2 and the pseudocode for the algorithm can be found in Appendix A.2. During training, the model can be delineated into two stages: the initial stage involves training the compressor and incorporating image branch controls, while the subsequent stage entails training the semantic pre-embedding module and integrating mixed semantic branch controls. During inference, the process can be dichotomized into image compression and image reconstruction. Within the image compression process, the input image $x$ is initially passed through the pre-trained encoder $\mathcal{E}$ to obtain $z_0 = \mathcal{E}(x)$. Simultaneously, intermediate features $f_1$ and $f_2$ are extracted. These outputs are then fed into the multi-feature compressor $M_\phi(\cdot)$, yielding distorted image-level control $\hat{c}$:

$$y = M_\phi^e(z_0, f_1, f_2), \ \hat{y} = Q(y), \ \hat{c} = M_\phi^d(\hat{y}). \tag{6}$$

$M_\phi^e$ and $M_\phi^d$ represent the encoder and decoder of the compressor $M_\phi$, respectively, with $Q$ denoting the quantization operation. Simultaneously, the textual description $text_x$ derived from image-captioning for $x$ undergoes lossless encoding and is transmitted alongside $\hat{c}$ to the decoding end. Notably, the transmission of $text_x$ occupies a minimal number of bits ($\leq 0.0001$bpp).

During the image reconstruction phase, $\hat{c}$ is initially decoded by the entropy decoder and then input into the Stable-Diffusion's decoder $\mathcal{D}$ to obtain a degraded representation $\hat{c}_x$ of $x$. Subsequently, $\hat{c}_x$ and $text_x$ are jointly fed into the Q-Former ($\text{QF}(\cdot, \cdot)$) to yield a mixed semantic output $s_x$:

$$\hat{c}_x = \mathcal{D}(\hat{c}), \ s_x = \text{QF}(\mathbf{E}_{img}(\hat{c}_x), text_x), \tag{7}$$

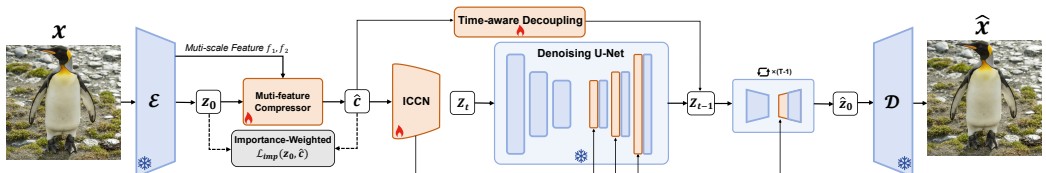

(a) First-stage: Learning image-level compressed representation

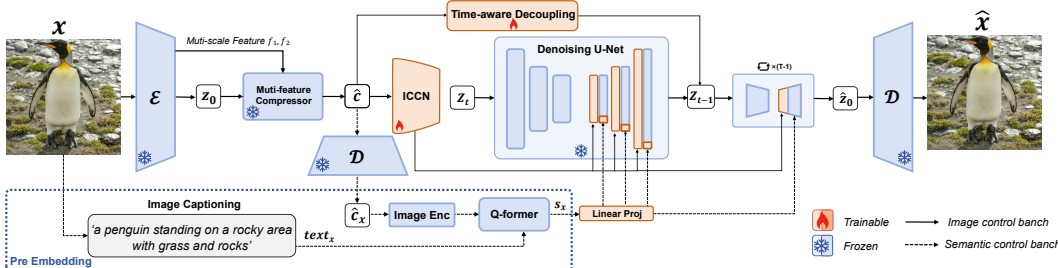

(b) Second-stage: Learning semantic-level hybrid condition

Figure 2: Illustration of the proposed Diffusion-based High Perceptual Fidelity Image Compression with Semantic Refinement (DiffPC). (a) In the first stage, DiffPC employs a variational compressor that receives multi-scale features to generate distorted image control branches. These low-level conditions are used to govern the pre-trained latent diffusion model through **IC-ControNet** (ICCN) (Section 3.2). (b) In the second stage, DiffPC utilizes a pre-embedding module to efficiently inject a blend of textual and visual semantics (Section 3.3).

where $\mathbf{E}_{img}(\cdot)$ stands for pre-trained image encoder. The semantic control flow $s_x$ and the image control flow $\hat{c}$ will individually exert control over the conditional diffusion model through cross-attention layers and the IC-ControlNet. Subsequently, after sampling and decoding, the reconstruction image $\hat{x}$ is obtained:

$$\hat{z}_0 = \text{Sampler}\left(z_t, \mathcal{DN}_\theta\left(\cdot, \hat{c}, s_x, t\right), T\right), \ \hat{x} = \mathcal{D}(\hat{z}_0). \tag{8}$$

Here, $T$ represents the number of sampling steps.

## 3.2 Stage I: Learning Compressor and low-level conditions

**Multi-feature Compressor** To simultaneously reduce the redundancy of latent $z_0$ and prevent information loss caused by downsampling in the encoder $\mathcal{E}$, we have devised a variational multi-feature compressor $M_\phi(\cdot)$, as depicted in the Figure 3(a). This compressor leverages residual extraction modules and cross-attention to fuse multi-scale features $f_1$ and $f_2$. We have adopted the SCCTX from paper (He et al., 2022) as the foundational entropy model. Diverging from neural compressors in the image domain, $M_\phi(\cdot)$ reconstructs the control terms of the conditional diffusion model. To ensure that the results of conditional generation align with expectations, we aim to minimize the following Kullback-Leibler divergence:

$$D_{\text{KL}}(p(\mathbf{z}_0|\mathbf{x}), p(\mathbf{z}_0|\hat{\mathbf{c}})) = D_{\text{KL}}(p(\mathbf{z}_0|\mathbf{x}), p(\mathbf{z}_0|M_\phi(\mathbf{z}_0, f_1, f_2))). \tag{9}$$

The encoder $\mathcal{E}$ of the pre-trained VAE used in Stable Diffusion explicitly models $p(\mathbf{z}_0|\mathbf{x})$ [2]. However, obtaining $p(\mathbf{z}_0|\hat{\mathbf{c}})$ during training comes at a high cost, requiring numerous samples to be taken during the diffusion process. Fortunately, we have demonstrated that optimizing the above objective does not necessitate frequent sampling during retraining:

**Theorem 3.1.** *Given the input image* $\mathbf{x}$*, the VAE-based encoder* $\mathcal{E}$*, the VAE-based compressor* $M_\phi$*, where* $\mathbf{z}_0 = \mathcal{E}(\mathbf{x})$*,* $\hat{\mathbf{c}} = M_\phi(\mathbf{z}_0)$*. We have:*

$$D_{\text{KL}}(p(\mathbf{z}_0|\mathbf{x}), p(\mathbf{z}_0|\hat{\mathbf{c}})) \leq D_{\text{KL}}(p(\mathbf{z}_0|\mathbf{x}), p_\gamma(\hat{\mathbf{c}}|\mathbf{z}_0)). \tag{10}$$

*Proof.* See Appendix A.1. □

---

[2]Here, $\mathbf{z}_0$ refers to the population of the sample vector $\mathbf{z}_0$, and so forth.

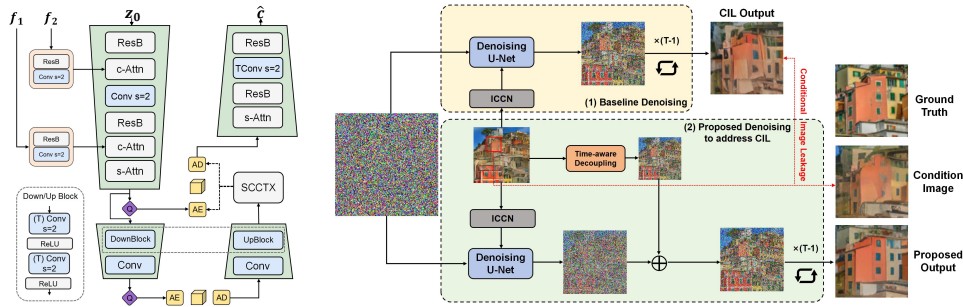

(a) Architecture of multi-feature compressor    (b) Illustration of Condition Leakage and Time-Aware Decoupling

Figure 3: Illustration of each module in the first stage. (a) The multi-feature compressor we propose takes multiple inputs $z_0$, $f_1$, and $f_2$. **s-Attn** and **c-Attn** denote self-attention and cross-attention, respectively. (b) During the baseline denoising process, Conditional Image Leakage (CIL) occurred, causing the output image (CIL Output) to mimic distortions and artifacts from conditional images. Our proposed solution circumvents this phenomenon.

Furthermore, assuming both distributions are parameterized normal distributions with equal variance, we have $p(\mathbf{z}_0|\mathbf{x}) = \mathcal{N}(\mathbf{z}; \boldsymbol{\mu}_{\mathbf{z}_0}, \boldsymbol{\Sigma})$ and $p_\gamma(\hat{\mathbf{c}}|\mathbf{z}_0) = \mathcal{N}(\hat{\mathbf{c}}; \boldsymbol{\mu}_{\hat{\mathbf{c}}}, \boldsymbol{\Sigma})$, where the covariance matrix $\boldsymbol{\Sigma} = \sigma_{\mathbf{z}_0}^2 \mathbf{I}$. In this scenario, the optimization objective transforms into a variance-weighted MSE:

$$\underset{\boldsymbol{\gamma}}{\arg\min}\, D_{\mathrm{KL}}(p(\mathbf{z}_0|\mathbf{x}), p_\gamma(\hat{\mathbf{c}}|\mathbf{z}_0)) = \underset{\boldsymbol{\gamma}}{\arg\min}\, \frac{1}{2\sigma_{\mathbf{z}_0}^2}\left[\|\boldsymbol{\mu}_{\mathbf{z}_0} - \boldsymbol{\mu}_{\hat{\mathbf{c}}}\|_2^2\right]. \tag{11}$$

In fact, $\sigma_{\mathbf{z}_0}$ models the importance of the image latent. Significant high-frequency regions (such as textures and edges) are modeled by $\mathcal{E}$ with lower variances, while less crucial flat regions (like large areas of uniform color) are modeled with higher variances. Following this intuitive observation, we set the variance as a trainable hyperparameter $w$, initialized to $\sigma_{\mathbf{z}_0}^2$:

$$\mathcal{L}_{imp}\left(\boldsymbol{\mu}_{\mathbf{z}_0}, \boldsymbol{\mu}_{\hat{\mathbf{c}}}\right) = \frac{1}{w}\mathcal{L}_{mse}\left(\boldsymbol{\mu}_{\mathbf{z}_0}, \boldsymbol{\mu}_{\hat{\mathbf{c}}}\right). \tag{12}$$

We refer to this optimization objective as Importance-Weighted MSE. The importance-weighted loss assigns more bits to texture details, enabling a more precise reconstruction of these features. We showcase visualizations of the bit rate allocation in Appendix A.7.3 to substantiate this point. In practice, the mean of $\mathbf{z}_0$, $\mathbf{c}$ are often reduced to their samples $\mathbf{z}_0, \hat{\mathbf{c}}$.

**IC-ControlNet and Time-aware Decoupling** In compression tasks, ControlNet provides robust non-semantic control for conditional diffusion models; however, it falls short in precision control, resulting in a significant decline in fidelity, as illustrated in Figure 8. To address this challenge, we propose the IC-ControlNet framework. During the input stage, we utilize convolutional modulation layers to effectively accommodate distorted conditions $\hat{\mathbf{c}}$. These conditions are then integrated with noise $\mathbf{z}_t$ and processed through residual blocks before entering the main network. IC-ControlNet enhances the control intensity of $\hat{\mathbf{c}}$, reducing uncertainties during the generation process.

Nevertheless, an excessively stringent control at low bitrates harbors drawbacks: aggressive quantization may induce pathological degradation in $\hat{\mathbf{c}}$, escalating the challenge of predicting $\mathbf{z}_0$ for the denoising model. This tendency may incline the denoising model to forsake forecasting the true data distribution and instead produce samples akin to the conditioned $\hat{\mathbf{c}}$; this phenomenon is termed as *condition leakage* (Zhao et al., 2024). Within the compression framework, this phenomenon manifests as the denoising model forfeiting its denoising efficacy, consequently reconstructing distorted images, as illustrated in Figure 3(b)(1). To address this issue, we propose the Time-Aware Decoupling (TAD) module. The details of TAD can be found in the Appendix A.4. TAD endeavors to disentangle the control factor $\hat{\mathbf{c}}$ from IC-ControlNet, thereby transforming the noise prediction of the diffusion model into residual noise prediction:

$$\hat{\epsilon} = \mathcal{DN}_\theta\left(\sqrt{\bar{\alpha}_t}\mathbf{z}_0 + \sqrt{1 - \bar{\alpha}_t}\epsilon, \hat{\mathbf{c}}, t\right) + \mathrm{TAD}_\eta(\hat{\mathbf{c}}, t). \tag{13}$$

$\eta$ represents the optimizable parameter of TAD. The incorporation of TAD and residual structures ensures the stability and reliability of the denoising process, as illustrated in Figure 3(b)(2). For

further experimental results, please refer to Section 4.3. In essence, the initial phase is supervised by:

$$\mathcal{L}_{stage\ 1} = \lambda_1 \mathcal{L}_{imp}(\boldsymbol{z}_0, \hat{\boldsymbol{c}}) + \lambda_2 \mathcal{L}_{rate} + \mathcal{L}_{CSD}, \tag{14}$$

$$\mathcal{L}_{CSD} = \mathbb{E}_{\boldsymbol{z}_0, \hat{\boldsymbol{c}}, t, \epsilon} \left[ \left\| \epsilon - \mathcal{DN}_\theta \left( \sqrt{\bar{\alpha}_t} \boldsymbol{z}_0 + \sqrt{1 - \bar{\alpha}_t} \epsilon, \hat{\boldsymbol{c}}, t \right) - \text{TAD}_\eta(\hat{\boldsymbol{c}}, t) \right\|_2^2 \right]. \tag{15}$$

Here, $\mathcal{L}_{rate}$ denotes the learned quantized latent representation by the compressor along with the hyperprior compression rate. The parameters $\lambda_1$ and $\lambda_2$ are utilized to achieve a trade-off between rate and distortion.

### 3.3 STAGE II: LEARNING SEMANTIC EMBEDDING AND HIGH-LEVEL CONDITIONS

Our experiments demonstrate that relying solely on low-level image control branches fails to achieve satisfactory realism. Additionally, the direct integration of high-level textual semantics through the stable diffusion interface proves ineffective. To overcome this challenge and avoid the costly iterative learning of semantic representations as highlighted in (Lei et al., 2023), we have developed a semantic pre-embedding module. As depicted in Figure 2, the decoding process begins by acquiring a degraded representation $\hat{\boldsymbol{c}}_x$. Although $\hat{\boldsymbol{c}}_x$ lacks significant texture and detail compared to $\boldsymbol{x}$, it retains essential low-frequency information, such as object shapes and colors—elements that are often absent in textual semantics. The visual semantics from $\hat{\boldsymbol{c}}_x$ and the textual semantics $text_x$ derived from image captioning are then jointly integrated into a pre-trained Q-Former, resulting in a blended semantic output $\boldsymbol{s}_x$.

During this phase, we keep the parameters of the compressor $M_\phi$ frozen. To enable the frozen conditional denoising network to adapt to the new semantic input, we have introduced a linear projection layer to modulate the blended semantic $\boldsymbol{s}_x$ and unfreeze all cross-attention layers integrating semantic fusion within the denoising diffusion network for fine-tuning. Experimental results demonstrate a notable enhancement in the perceptual quality of decoded images through the fine-tuning of attention layers and semantic pre-embedding.

The loss in the second phase is solely diffusion loss:

$$\mathcal{L}_{stage\ 2} = \mathbb{E}_{\boldsymbol{z}_0, \hat{\boldsymbol{c}}, \boldsymbol{s}_x, t, \epsilon} \left[ \left\| \epsilon - \mathcal{DN}_\theta \left( \sqrt{\bar{\alpha}_t} \boldsymbol{z}_0 + \sqrt{1 - \bar{\alpha}_t} \epsilon, \hat{\boldsymbol{c}}, \boldsymbol{s}_x, t \right) - \text{TAD}_\eta(\hat{\boldsymbol{c}}, t) \right\|_2^2 \right]. \tag{16}$$

To address the color shift issue caused by the diffusion model (Choi et al., 2022), we were inspired by (Wang et al., 2024) to perform *color correction* on the decoded image $\hat{\boldsymbol{x}}$. Interestingly, we observed that the colors in degraded representation $\hat{\boldsymbol{c}}_x$ retain accuracy even as it lose a substantial amount of high-frequency components. Consequently, we normalize the color of the decoded image $\hat{\boldsymbol{x}}$ to align its mean and variance with $\hat{\boldsymbol{c}}_x$. Further, we have found that color correction can be achieved through a learnable decoder that enhances certain perceptual metrics. For more details, please refer to the Appendix A.4.

## 4 EXPERIMENTS

### 4.1 EXPERIMENTAL SETUP

**Metrics.** Our statistical metrics can be categorized into the following three groups. *(1) Reference-based distortion-based metrics*: PSNR and MS-SSIM (Wang et al., 2003). These metrics offer pixel-level distortions but have been shown to be ineffective at providing a valid measure of perceptual realism at low bit rates. *(2) Reference-based perceptual metrics*: LPIPS (Zhang et al., 2018), DISTS (Ding et al., 2020). These metrics are widely utilized for assessing the perceptual quality of images and have been demonstrated to be more correlated with human judgment. *(3) No-reference perceptual metrics*: FID (Heusel et al., 2017), KID (Bińkowski et al., 2018), CLIP-IQA (Wang et al., 2023). Among these, FID and KID measure the distribution difference between compressed and original images, capturing the statistical fidelity of the compression scheme. CLIP-IQA is an image quality assessment metric that utilizes a cross-modal model to score the realism and perceptual quality of images. **Baseline.** We compared DiffPC with several state-of-the-art neural compression schemes. This includes ELIC (He et al., 2022), a model based on VAE and rate-distortion optimization, HiFiC (Mentzer et al., 2020) based on GAN architecture and perceptual loss, and its enhanced version, MS-ILLM (Muckley et al., 2023). Furthermore, we also compared our approach with VQGAN based

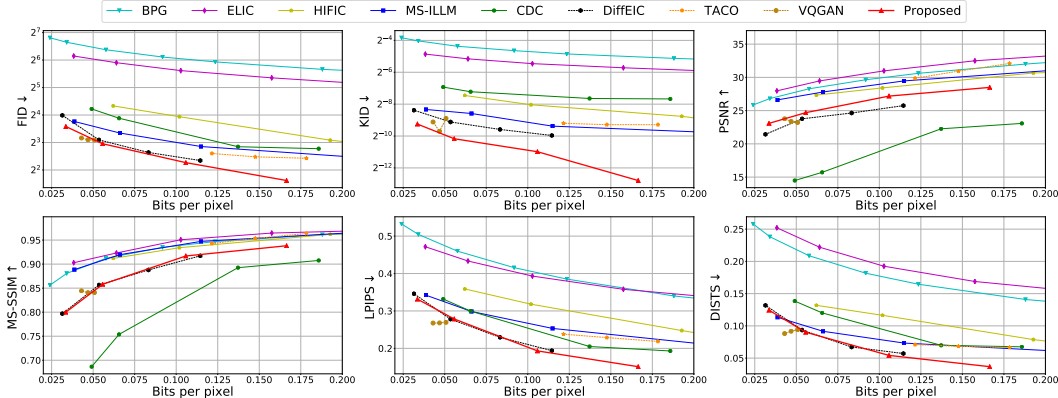

Figure 4: Comparisons of methods across various metrics for the CLIC 2020 test set.

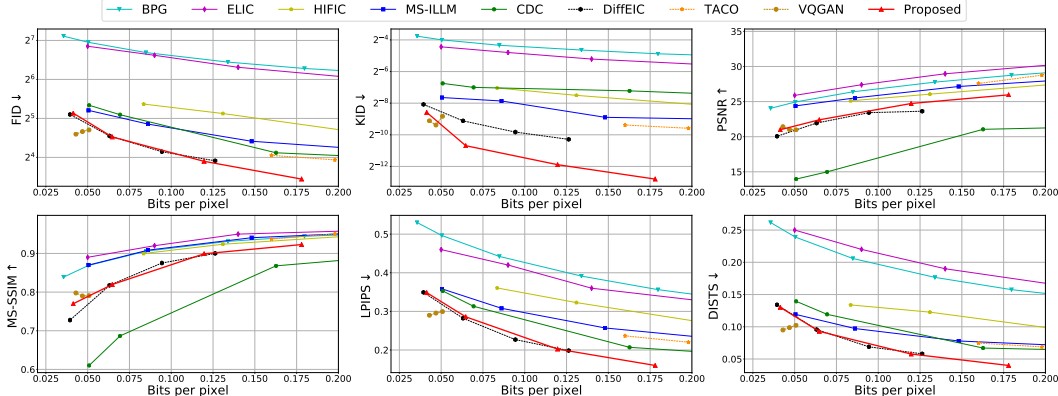

Figure 5: Comparisons of methods across various metrics for the DIV2K validation set.

(Mao et al., 2024) methods and the solution TACO (Lee et al., 2024) that incorporates textual semantic priors. For the diffusion model-based compression baseline, we conducted our main experiments comparing against CDC (Yang & Mandt, 2024) and DiffEIC Li et al. (2024b). To ensure a fair comparison, we retraced and retrained these baselines on the LSDIR dataset using their open-source code and default settings, resulting in training outcomes closely aligned with the original results. It is worth noting that for VQGAN (Mao et al., 2024), we utilized the official checkpoint trained on the ImageNet (Deng et al., 2009) dataset, which is 14 times larger than the LSDIR dataset.

In the case of HiFiC and CDC, we not only utilized their reported lowest bit rates but also extended the comparison to even lower bit rates. Additionally, we contrasted these approaches with the traditional compression format BPG (Bellard), which is the image compression component of HEVC. It is noteworthy that due to some diffusion-based baselines (Hoogeboom et al., 2023; Careil et al., 2024) lacking sufficient validation results and experimental details, we only present DiffPC's superior statistical fidelity in the Appendix A.8.

**Datasets.** For validation, we referenced (Muckley et al., 2023) and employed three widely recognized image compression benchmark datasets: CLIC2020 (George Toderici, 2020), DIV2K (Timofte et al., 2017), and Kodak (Company). CLIC2020 comprises 428 high-definition images, DIV2K includes 100 2K resolution high-definition images, and Kodak consists of 24 natural images with a resolution of $768 \times 512$. Due to computational constraints, except for the Kodak dataset, we center-cropped the original images to $1024 \times 1024$ resolution. As Kodak lacks extensive validation, we present the experimental results in the Appendix A.8.

Furthermore, following the approaches of (Hoogeboom et al., 2023; Careil et al., 2024), we validated the model's statistical fidelity using COCO30K (Lin et al., 2014) and present the results in the Appendix A.8.

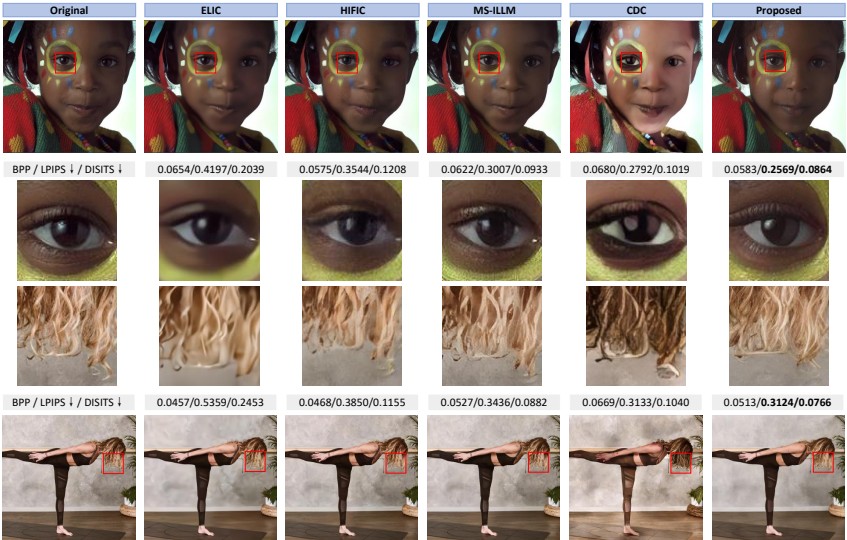

Figure 6: Qualitative illustrations of various methods on Kodak and CLIC2020 datasets. DiffPC reconstructs images without any artifacts and delicately restores intricate and complex textures.

## 4.2 MAIN RESULTS

**Quantitative comparisons.**

Figures 4 and 5 shows the performance across different bit rates. ELIC, based on rate-distortion optimization, exhibits significant shortcomings across all perceptual metrics, failing to outperform even the hand-craft image compression standard BPG. HiFiC maintains excellent performance at higher bit rates; however, it shows poorer performance at lower bit rates ($\leq 0.1$ bpp). MS-ILLM demonstrates a satisfactory performance improvement over HiFiC at lower bit rates, approaching CDC's performance. Never-

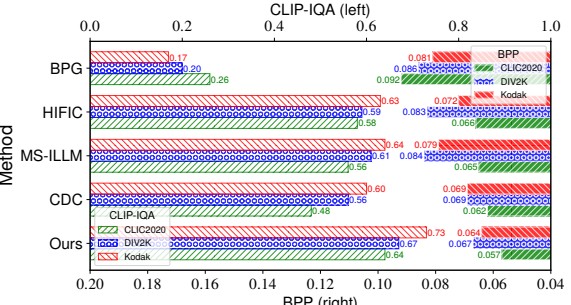

Figure 7: Comparing visual quality using CLIP-IQA.

theless, our proposed DiffPC consistently outperforms all baseline solutions in perceptual quality across all bit rates. Additionally, DiffPC achieves significantly better performance in the FID and KID metrics compared to other baselines, indicating that our approach maintains excellent statistical fidelity even at low bitrates. Compared to TACO, which also utilizes textual semantics, DiffPC achieved significantly higher perceptual fidelity and statistical fidelity in low bitrate scenarios. In contrast to DiffEIC, which employs a pre-trained diffusion model prior, DiffPC attained superior statistical fidelity through the fusion of mixed semantics. Additionally, due to improved bitrate allocation, our approach exhibits lower distortion and enhanced perceptual fidelity compared to DiffEIC. Further, we observe that the generalization performance of VQ-GAN in terms of bitrates is concerning: it exhibits an anomalous situation where the rate-distortion curve shows a monotonous increase at boundary bitrates. In contrast, DiffPC demonstrates stronger bitrate generalization capabilities and the ability to adapt to various fidelity requirements. Besides, we compared pixel-level distortions, although they are proven to have limited reference value at lower bitrates (Careil et al., 2024). DiffPC demonstrates better fidelity in distortion metrics compared to the diffusion-based CDC and competes with HiFiC in performance.

**Qualitative comparisons.** We compared the visual quality of image reconstruction between DiffPC and baseline methods at ultra-low bit rates ($\leq 0.08$ bpp) using CLIP-IQA, as shown in Figure 7. Across three datasets, DiffPC reconstructed images at the lowest bit rate that significantly surpassed

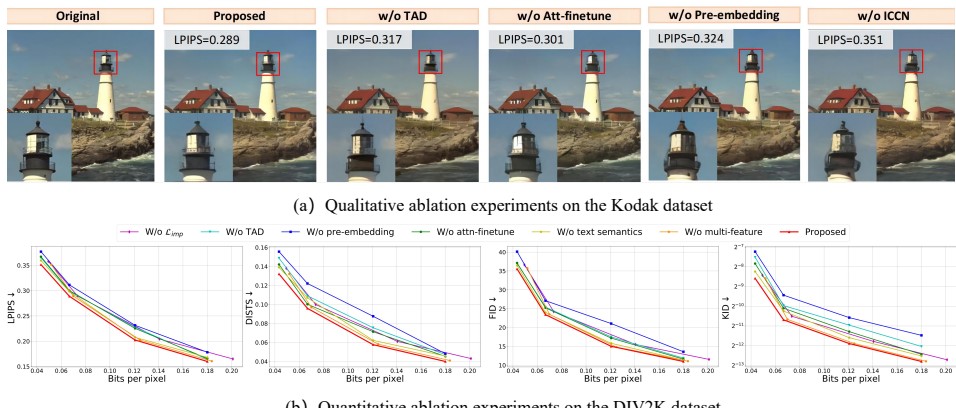

(a) Qualitative ablation experiments on the Kodak dataset

(b) Quantitative ablation experiments on the DIV2K dataset

Figure 8: Ablation study of the modules in DiffPC.

other neural codecs in realism. In Figures 1 and 6, we presented visualizations of the reconstructed images. At ultra-low bit rates, while ELIC's reconstructed images exhibit minimal distortion, their visual quality is dismal. HiFiC and MS-ILLM's optimization relies on a single perceptual metric, leading to varying degrees of artifacts and noise at low bit rates. CDC, constrained by the DDPM structure, exhibits certain color shifts and detail distortions during image decoding. Conversely, DiffPC's generated results at low bit rates demonstrate remarkably superior visual quality and realism. Whether capturing intricate textures (such as hair in Figure 6 and foliage in Figure 1) or finer details (like eyelashes in Figure 6), DiffPC excels in capturing and reliably reconstructing with minimal bitrates.

## 4.3 ABLATION STUDY

In this section, we conducted ablation studies on various modules of DiffPC, and the quantitative and qualitative results are depicted in Figure 8.

*(1) W/O Importance-Weighted MSE*: By substituting $\mathcal{L}_{imp}$ with a standard MSE, as shown in Figure 8(b), the model fails to achieve the perceptual quality of the original design even with increased bit rates. *(2) w/o TAD*: Removing the TAD module significantly impairs the model's performance at extremely low bit rates and alters the object morphology in the generated images, as illustrated in Figure 8(a)(b). *(3) w/o ICCN*: Directly replacing IC-ControlNet with ControlNet, as seen in Figure 8(a), results in almost ineffective control within the diffusion framework. *(4) w/o Pre-embedding*: By omitting Q-Former and solely relying on textual descriptions for semantic control, there is a notable decrease in visual quality. Injecting simple text semantics in a conventional manner leads to a similar outcome as injecting noise. *(5) w/o Attn-finetune*: By not fine-tuning the cross-attention layers in the second training stage, the difficulty of integrating semantic control branches into the denoising network increases, directly causing performance degradation. *(6) w/o multi-feature*: The removal of the multiscale feature structure from the compressor resulted in the loss of high-frequency information. This leads to a significant performance drop at ultra-low bit rates. *(7) w/o text semantics*: We substituted the text prompts in the pre-embedding module with empty strings. Due to the lack of textual cues aiding global semantics and stability, performance shows a noticeable decline at ultra-low bit rates.

## 5 CONCLUSION

We introduce a novel neural compression framework, DiffPC, which leverages the priors from a pre-trained latent diffusion model to reconstruct images with high realism and visual quality in low bitrate scenarios. Unlike other generative model-based compression schemes, DiffPC achieves precise bit-rate allocation through a multi-feature compressor and incorporates a pre-embedding module for efficient semantic information injection into the conditional diffusion model. Extensive experiments demonstrate that our proposed approach faithfully reconstructs images even at extremely low bit rates while preserving high perceptual quality textures.

## 6  ACKNOWLEDGEMENTS

This work is supported in part by the National Natural Science Foundation of China under grant 62301189, 624B2088, the PCNL KEY project (PCL2023AS6-1), and Shenzhen Science and Technology Program under Grant KJZD20240903103702004, JCYJ20220818101012025, RCBS20221008093124061, GXWD20220811172936001.

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

# A APPENDIX

## A.1 PROOFS

**Theorem A.1.** *Given the input image* $\mathbf{x}$*, the VAE-based encoder* $\mathcal{E}$*, the VAE-based compressor* $M_\phi$*, and the compressor's encoder* $M_\phi^e$*,the compressor's decoder* $M_\phi^e$*, Quantization operation* $Q$*, where* $\mathbf{z}_0 = \mathcal{E}(\mathbf{x})$*,* $\mathbf{y} = M_\phi^e(\mathbf{z}_0)$*,* $\hat{\mathbf{y}} = Q(\mathbf{y})$*,* $\hat{\mathbf{c}} = M_\phi^d(\hat{\mathbf{y}})$*. We have:*

$$D_{\mathrm{KL}}(p(\mathbf{z}_0|\mathbf{x}), p(\mathbf{z}_0|\hat{\mathbf{c}})) \leq D_{\mathrm{KL}}(p(\mathbf{z}_0|\mathbf{x}), p_\gamma(\hat{\mathbf{c}}|\mathbf{z}_0)). \tag{17}$$

*Proof.* According to the definition of KL divergence, we have:

$$D_{\mathrm{KL}}(p(\mathbf{z}_0|\mathbf{x}), p(\mathbf{z}_0|\hat{\mathbf{c}})) = \int p(\mathbf{z}_0|\mathbf{x}) \log \frac{p(\mathbf{z}_0|\mathbf{x})p(\hat{\mathbf{c}})}{p_\gamma(\hat{\mathbf{c}}|\mathbf{z}_0)p(\mathbf{z}_0)} d\mathbf{z}_0 \tag{18}$$

$$= D_{\mathrm{KL}}(p(\mathbf{z}_0|\mathbf{x}), p_\gamma(\hat{\mathbf{c}}|\mathbf{z}_0)) + \int p(\mathbf{z}_0|\mathbf{x}) \log \frac{p(\hat{\mathbf{c}})}{p(\mathbf{z}_0)} d\mathbf{z}_0. \tag{19}$$

Since $M_\phi$ is VAE-based and $\hat{\mathbf{c}} = M_\phi(\mathbf{z}_0)$. According to (Kingma, 2013), $\hat{\mathbf{c}}$ is generated by $M_\phi^d(\hat{\mathbf{y}})$. Due to the distortion caused by quantization $Q$, $M_\phi^d(\hat{\mathbf{y}})$ actually estimates the distribution $p(\mathbf{z}_0|\hat{\mathbf{y}})$. Therefore, we have:

$$D_{\mathrm{KL}}(p(\mathbf{z}_0|\mathbf{x}), p(\mathbf{z}_0|\hat{\mathbf{c}})) = D_{\mathrm{KL}}(p(\mathbf{z}_0|\mathbf{x}), p_\gamma(\hat{\mathbf{c}}|\mathbf{z}_0)) + \int p(\mathbf{z}_0|\mathbf{x}) \log \frac{p(\mathbf{z}_0|\hat{\mathbf{y}})}{p(\mathbf{z}_0)} d\mathbf{z}_0 \tag{20}$$

$$= D_{\mathrm{KL}}(p(\mathbf{z}_0|\mathbf{x}), p_\gamma(\hat{\mathbf{c}}|\mathbf{z}_0)) + \int p(\mathbf{z}_0|\mathbf{x}) \log \frac{p(\mathbf{z}_0|\hat{\mathbf{y}})}{\int p(\mathbf{z}_0|\hat{\mathbf{y}})d\hat{\mathbf{y}}} d\mathbf{z}_0 \tag{21}$$

$$< D_{\mathrm{KL}}(p(\mathbf{z}_0|\mathbf{x}), p_\gamma(\hat{\mathbf{c}}|\mathbf{z}_0)). \tag{22}$$

When $Q$ is the identity function (meaning there is no quantization distortion), in this case, $M_\phi^d(\hat{\mathbf{y}}) = M_\phi^d(\mathbf{y})$ fits the input distribution $p(\mathbf{z}_0)$, so we have:

$$D_{\mathrm{KL}}(p(\mathbf{z}_0|\mathbf{x}), p_\gamma(\hat{\mathbf{c}}|\mathbf{z}_0)) + \int p(\mathbf{z}_0|\mathbf{x}) \log \frac{p(\hat{\mathbf{c}})}{p(\mathbf{z}_0)} d\mathbf{z}_0 = D_{\mathrm{KL}}(p(\mathbf{z}_0|\mathbf{x}), p_\gamma(\hat{\mathbf{c}}|\mathbf{z}_0)). \tag{23}$$

Above all:

$$D_{\mathrm{KL}}(p(\mathbf{z}_0|\mathbf{x}), p(\mathbf{z}_0|\hat{\mathbf{c}})) \leq D_{\mathrm{KL}}(p(\mathbf{z}_0|\mathbf{x}), p_\gamma(\hat{\mathbf{c}}|\mathbf{z}_0)). \tag{24}$$

$\square$

## A.2 MORE DETAILS OF ALGORITHM PROCEDURE

In this section, we supplement the explanation of the encoding and decoding processes of DiffPC through pseudocode. The detailed encoding and decoding processes are shown below.

## A.3 FURTHER EXPERIMENTAL DETAILS

### A.3.1 DETAILS OF THE MODEL TRAINING

Our model was trained on the LSDIR dataset (Li et al., 2023b), which comprises 84,991 high-definition natural images. During training, these images were randomly cropped to a resolution of $512 \times 512$. Our foundational conditional diffusion model leverages Stable Diffusion 2.1-base[3].. Throughout all training stages, we employed AdamW (Loshchilov, 2017) as the optimizer, with learning rates set at $1 \times 10^{-4}$ for the initial phase and $5 \times 10^{-5}$ for the subsequent phase. The batch size was consistently maintained at 2.

In the initial training phase, we employed an entropy estimator SCCTX (He et al., 2022) with a group number of 3. To achieve compression at different bit rates, we set the parameter $\lambda_2$ in Section 3.2 to 0.2 and then adjusted $\lambda_1 \in \{4, 16, 64, 128\}$. At this stage, we will train with 80000 steps.

---

[3]https://huggingface.co/stabilityai/stable-diffusion-2-1-base

---

**Algorithm 1** Encoding Process

---

1: Given input data $\boldsymbol{x}$, compressor encoder $M_\phi^e(\cdot)$, stable diffusion's encoder $\mathcal{E}(\cdot)$, quantization operation $Q$
2: $\boldsymbol{z}_0, f_1, f_2 = \mathcal{E}(\boldsymbol{x})$
3: $\boldsymbol{y} = M_\phi^e(\boldsymbol{z}_0, f_1, f_2)$
4: $\hat{\boldsymbol{y}} = Q(\boldsymbol{y})$
5: $text_x = \text{Image\_captioning}(\boldsymbol{x})$
6: Encode $\hat{\boldsymbol{y}}$, $text_x$ to binary file
7: Output encoded data

---

**Algorithm 2** Decoding Process

---

1: Given compressor decoder $M_\phi^d(\cdot)$, stable diffusion's decoder $\mathcal{D}(\cdot)$, pre-embedding module QF$(\cdot)$, Denosing network $\mathcal{DN}_\theta$, diffusion steps $T$
2: Decode $\hat{\boldsymbol{y}}$, $text_x$ from binary file
3: $\hat{\boldsymbol{c}} = M_\phi^d(\hat{\boldsymbol{y}})$
4: $\hat{\boldsymbol{c}}_x = \mathcal{D}(\hat{\boldsymbol{c}})$
5: $\boldsymbol{s}_x = \text{QF}(\hat{\boldsymbol{c}}_x, text_x)$
6: $\boldsymbol{z}_T \sim \mathcal{N}(0, \mathbf{I})$
7: **for** t $\in [T, \cdots, 1]$ **do**
8: $\quad \boldsymbol{z}_{t-1} = \text{Sampler}\left(\mathcal{DN}_\theta\left(\boldsymbol{z}_t, \hat{\boldsymbol{c}}, \boldsymbol{s}_x, t\right), t\right)$
9: **end for**
10: $\hat{\boldsymbol{x}} = \mathcal{D}(\boldsymbol{z}_0)$
11: $\boldsymbol{x}_{rec} = \text{Color\_correction}(\hat{\boldsymbol{x}}, \hat{\boldsymbol{c}}_x)$
12: Output $\boldsymbol{x}_{rec}$

---

In the second training phase, the parameters of the compressor were frozen. We utilized BLIP-2 (Li et al., 2023a) for extracting textual semantic descriptions of images and employed the pre-trained Q-Former and image-encoder from BLIP-diffusion (Li et al., 2024a). Finally, for color correction in the decoded output, we applied wavelet-color correction (Wang et al., 2024). At this stage, we will train with 60000 steps.

We did not apply warm-up in the first stage but utilized a LambdaLinearScheduler with parameters *warm_up_steps=10000* and *f_start=1e-6* in the second stage. For sampling, we utilized IDDPM (Nichol & Dhariwal, 2021) as the sampler with a uniform setting of 50 sampling steps, as reported to be optimal in the original paper. Additionally, all experiments were conducted on an Nvidia A6000 GPU. Code is released at https://github.com/Darc8-sun/DIFFPC.

### A.3.2 Details of the model testing

**Implementation of the baseline approach** All the baseline approaches we utilized employed their respective official open-source codes [4] [5] [6]. To ensure result alignment, these baselines were retrained on LSDIR. The results we reproduced closely match the original papers, with the retrained HiFiC even outperforming the performance of the official checkpoint during testing.

**Evaluation metrics for testing** For LPIPS, we utilized the *lpips* library, while DISTS was implemented using *DISTS_pytorch*. FID and KID metrics were calculated using functions provided by *torchmetrics.image*, with a feature size of 2048. Following the approach in (Mentzer et al., 2020), during FID and KID testing, images were partitioned. Specifically, from each $H \times W$ image, we initially extracted $\lfloor H/f \rfloor \cdot \lfloor W/f \rfloor$ non-overlapping $f \times f$ crops. Subsequently, the extraction origin was shifted by $f/2$ in both dimensions to obtain another $(\lfloor H/f \rfloor - 1) \cdot (\lfloor W/f \rfloor - 1)$ patches. A fixed value of $f = 256$ was used for all evaluations.

---

[4] CDC: https://github.com/buggyyang/CDC_compression

[5] HiFiC: https://github.com/Justin-Tan/high-fidelity-generative-compression

[6] MS-ILLM: https://github.com/facebookresearch/NeuralCompression

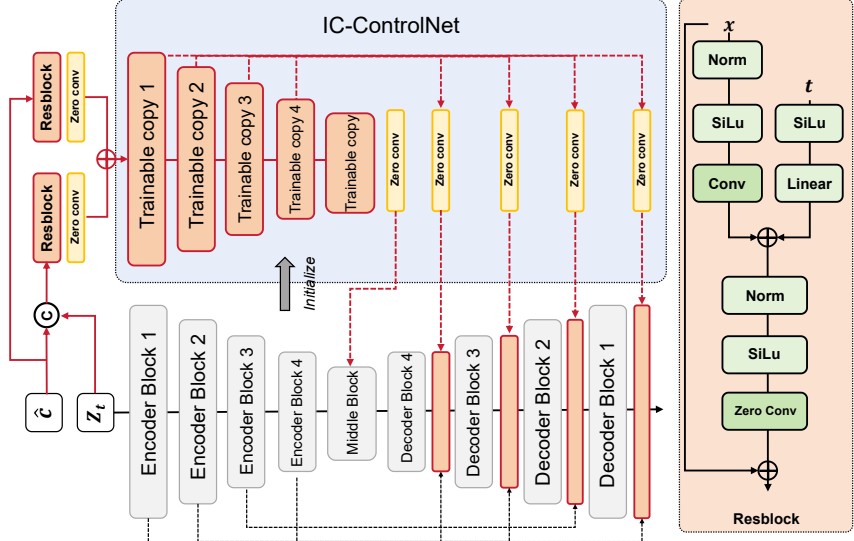

Figure 9: Architecture of IC-ControlNet.

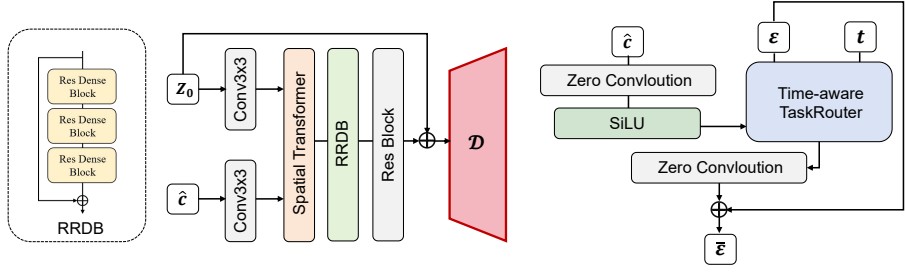

(a) Trainable decoders that can be used for color correction          (b) Architecture for Time-aware decoupling

Figure 10: Architecture of different modules.

## A.4 ARCHITECTURES OF MODULES

We intricately illustrated the structure of the control module IC-ControlNet and the architecture of the Time-aware decoupling designed in the first stage, as depicted in Figure 9 and Figure 10(b), respectively. The concept behind ICCN's design was inspired by (Lin et al., 2023), while the time-aware task router in the Time-aware decoupling was adapted from (Park et al., 2023). We treated each time step as an independent task, generating tasks to integrate temporal information.

As mentioned in Section 3.3, $\hat{c}$ possesses comprehensive color information, prompting us to devise a *trainable fusion decoder* for integration, as illustrated in Figure 10(a). This module comprises a feature fusion module and a trainable SD decoder $\mathcal{D}$. Since training this module occurs after the second-stage training, we term it the *3rd-stage training*. During the training of this decoder, all other parameters of the backbone network are frozen, and the LPIPS between the decoded image and the original image is employed as the loss for optimization. Our experiments indicate that this boosts perceptual metrics further during testing; however, conversely, statistical fidelity may marginally decrease, as shown in the Figure 12. Furthermore, akin to HiFiC and ILLM, we observed that optimizing with LPIPS inevitably introduces subtle grid-like artifacts. These artifacts primarily concentrate in intricate texture regions, as depicted in Figure 11. While this approach shows enhancements in some metrics, we believe it compromises the realism of the images, so it is presented solely as an optional additional component.

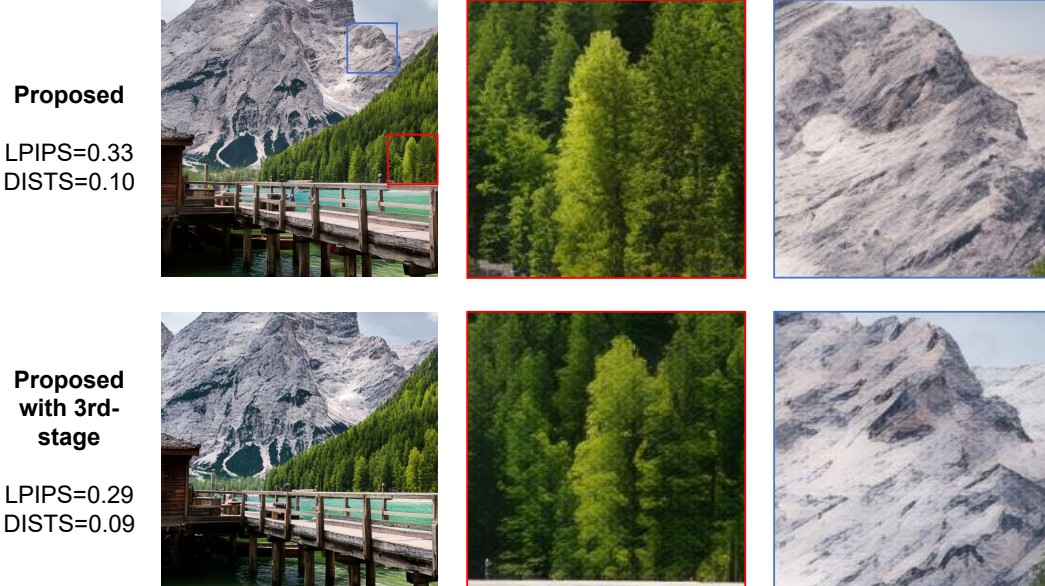

Figure 11: Qualitative ablation study of trainable fusion decoder.

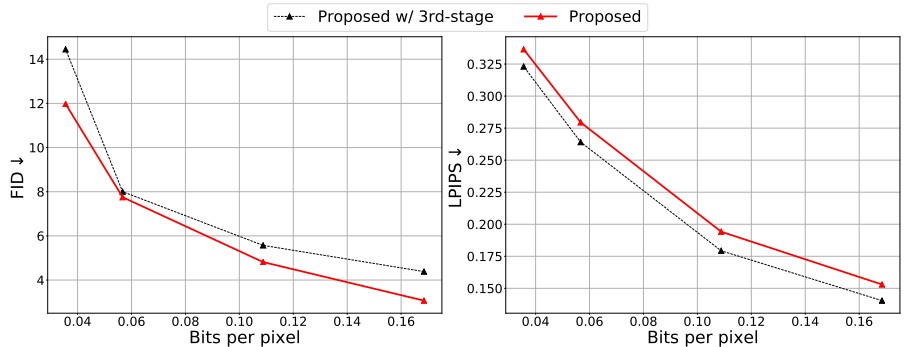

Figure 12: Quantitative ablation study of trainable fusion decoder.

### A.5 COMPLEXITY COMPARISONS

We compared the encoding and decoding latency of our models with the baseline in a GPU environment and included various performance comparisons on the rightmost side of the table, as shown in the Table 1. In the table, 'steps' represent the number of sampling steps required for decoding, with a default of 50 steps in our work. Our BD-rate is calculated with DiffPC as the baseline, where a higher value indicates a larger performance gap compared to DiffPC. It is important to note that, except for Perco, all tests were conducted on the same Nvidia 3080ti GPU. Due to high VRAM usage during inference, Perco was tested on an A6000 GPU, which has superior GFLOPS. In the table, we use the symbol "≥" to indicate that in the same GPU environment, Perco's encoding and decoding speeds exceed the values we provide.

It can be observed that by encoding only the latent code at the encoding end, DiffPC achieves satisfactory encoding speed: it outperforms most diffusion baselines in encoding speed and has a similar encoding latency to GAN-based methods Mentzer et al. (2020); Muckley et al. (2023). At the decoding end, we can actually reduce the number of sampling steps to decrease decoding latency. Even when reducing the sampling steps to 5, DiffPC maintains superior performance with considerable decoding speed. Even with 50 sampling steps, DiffEIC's decoding speed surpasses Perco, which is also based on a diffusion structure, and is comparable to the decoding speed of Cheng20, a neural compressor based on a sequence autoregressive entropy model.

| Model | Encoding (s) | Decoding (s) | BD-rate(%) |
|---|---|---|---|
| ELIC He et al. (2022) | 0.009 | 0.008 | 4254.39 |
| Cheng20 Cheng et al. (2020) | 3.081 | 6.678 | - |
| HiFiC Mentzer et al. (2020) | 0.013 | $3 \times 10^{-5}$ | 289.26 |
| MS-ILLM Muckley et al. (2023) | 0.069 | 0.068 | 79.83 |
| CDC Yang & Mandt (2024) | 0.007 | 3.080 | 143.54 |
| VQGAN based Mao et al. (2024) | 0.011 | 0.011 | - |
| TACO Lee et al. (2024) | 0.120 | 0.144 | 62.96 |
| DiffEIC Li et al. (2024b) | 0.430 | 6.619 | 10.45 |
| Perco Careil et al. (2024) | $\geq$0.767 | $\geq$15.261 | - |
| **DiffPC (steps=50)** | 0.089 | 7.325 | **0.00** |
| **DiffPC (steps=20)** | 0.089 | 3.378 | 6.25 |
| **DiffPC (steps=5)** | 0.089 | 0.886 | 14.59 |

Table 1: Encoding and decoding time (seconds) on Kodak dataset. BD-rate (%) is calculated on CLIC2020 dataset, with FID as the metric.

## A.6 SPECIAL SCENARIO DISCUSSION

In this section, we conducted a qualitative visual analysis of images containing 'text' and images containing 'faces'. Initially, we categorized 'text' and 'faces' into three classes based on their proportions within the entire frame - large, medium, and small - and discussed them separately, as shown in Figures 13 and 14.

In instances of small text (Figure 13), it is evident that even the most cutting-edge neural compressor ELIC based on traditional rate-distortion optimization cannot restore discernible text. Similarly, Gan-based methods also fall short. Given the premise of semantic loss in decoding, our proposed DiffPC decoding yields sharper edges and enhanced perceptual quality in the images. In the medium text category, all generative compression schemes exhibit some degree of text distortion. However, notably, compared to GAN-based methods which show more pronounced artifacts and structural deficiencies, DiffPC maintains the semantic consistency of text to the maximum extent while preserving details and textures in other parts. In the case of large text, all approaches can reconstruct high-fidelity textual information.

Regarding images containing faces (Figure 14), in the large and medium categories, DiffPC maintains remarkable structural consistency without distortions. Conversely, Gan-based solutions exhibit structural distortions, while ELIC shows severe blurring, making facial recognition challenging. In the small face category, both Gan-based methods and ELIC display significant distortions and structural chaos. In contrast, DiffPC sacrifices a certain level of semantic consistency to enhance overall structural coherence and texture details.

In conclusion, we believe that the proposed DiffPC achieves an optimal triple balance between realism, distortion, and bit rate at low encoding rates. Nonetheless, our approach still has shortcomings in maintaining semantic consistency for extremely small faces and text, which we aim to improve in future endeavors.

## A.7 FURTHER ABLATION EXPERIMENT

### A.7.1 NUMBER OF TIMESTEPS

In Figure 15, we show how DISTS, LPIPS, MS-SSIM and FID vary when we change the number of denoising steps. We do the evaluation for four different bitrates, ranging from 0.0043 to 0.1811 bpp. It is evident that at higher bit rates, reducing the number of samples has a minimal impact on performance degradation. This implies that in high bit rate scenarios, we can decrease sampling to accelerate decoding speed. At lower bit rates, perceptual metrics exhibit a more noticeable de-

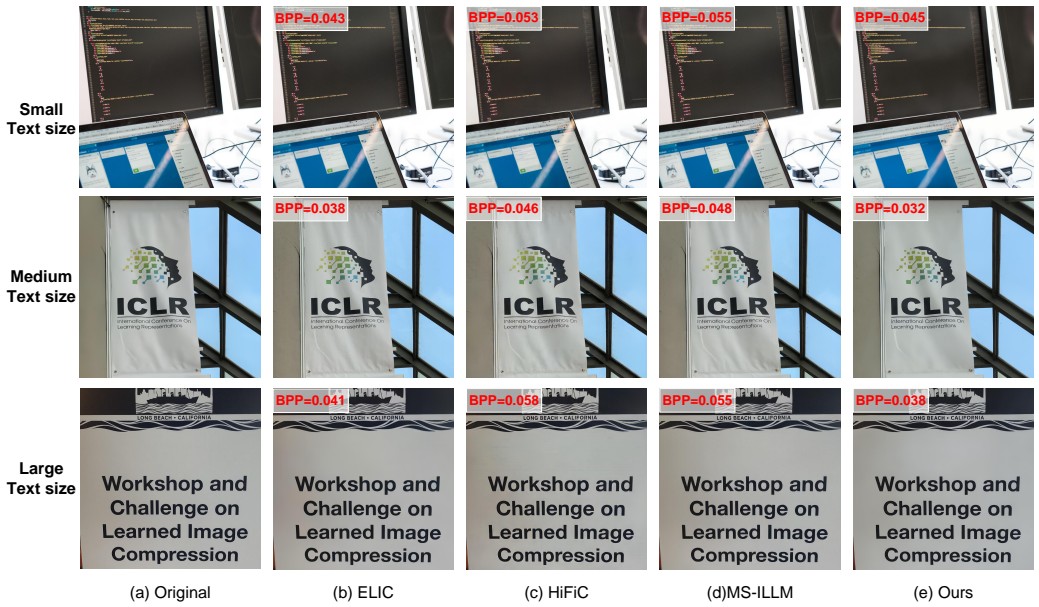

Figure 13: An image containing text. The image is from the CLIC2020 dataset, and the upper left corner of the image is the bit rate (BPP) of the image.

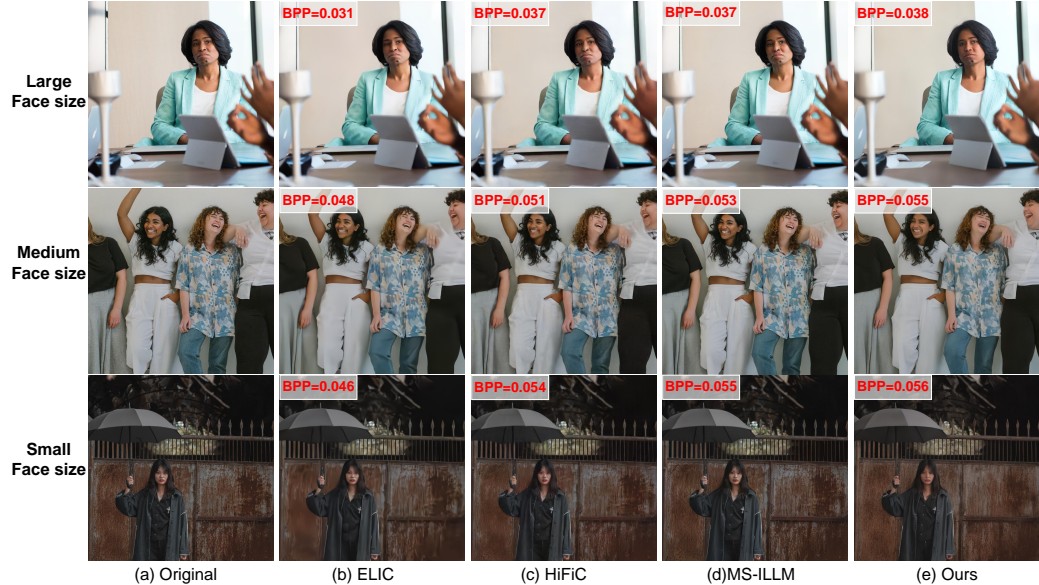

Figure 14: An image containing human face. The image is from the CLIC2020 dataset, and the upper left corner of the image is the bit rate (BPP) of the image.

cline with a reduction in the number of samples. However, overall, the statistical fidelity is not significantly affected by the number of samples.

### A.7.2 TEXT ROBUSTNESS

We employed four different approaches for textual descriptions: *(1) BLIP2 (5 ≤word count ≤ 10):* Forcing the word count output of BLIP2 to be within 10 words. *(2) OFA-tiny:* Using OFA (Wang et al., 2022) as an image-captioning model to obtain textual prompts for the images. *(3) Default text:* All images uniformly utilized the default text description: 'high quality, extreme detail.' *(4)*

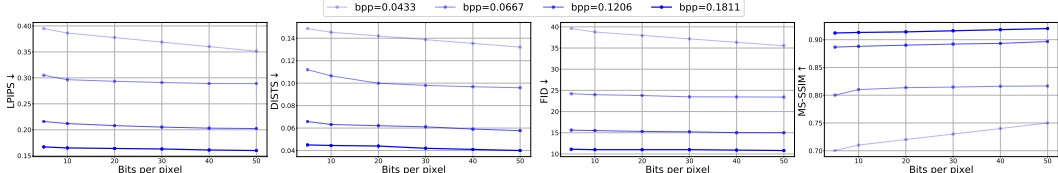

Figure 15: Quantitative comparisons of different number of denoising steps. Evaluation on DIV2K datasets.

| Text type | LPIPS | DISTS | PSNR |
|---|---|---|---|
| BLIP2 (5≤words count≤10) | 1.686 | 1.579 | 0.891 |
| OFA-tiny | 1.742 | -0.311 | 0.320 |
| Default text | 4.502 | 5.455 | 0.447 |
| Random string | 19.577 | 35.816 | 82.989 |
| **BLIP2 (10≤words count≤30)** | **0** | **0** | **0** |

Table 2: Text robustness evaluation. Tested on Kodak with a BD-rate (%) based on DiffPC (Ours). The bold sections represent the textual prompt acquisition approach used by DiffPC.

*Random string:* Employing random strings of equal length (20 characters). The result is shown in Appendix A.7.2. It is worth noting that the approach utilized by DiffPC falls under BLIP2 (10≤word count≤20), which serves as the baseline for our BD-rate calculations. It can be observed that as long as the textual description aligns with the semantic content of the image, it generally does not compromise the model's reconstruction performance. Even when default text prompts are used, the model can still derive benefits from the text.

### A.7.3 VISUALIZATION OF BIT ALLOCATION MAP

As mentioned in the Section 3.2, we employed importance-weighted loss to achieve improved bit rate allocation. We visualized partial latent representation bit rate allocation diagrams on the Kodak dataset, as depicted in Figure 16.

It is quite evident that the importance-weighted loss we employed effectively prevents the excessive allocation of bits in flat regions (such as the sky in the image) and instead allocates more bits to areas with intricate textures, which aligns with our expectations.

### A.8 FURTHER EXPERIMENTAL RESULTS

**Further Quantitative Results:** We present the results of all baseline methods along with DiffPC on the Kodak dataset, as shown in Figure 17. Due to the limited size of the Kodak dataset, which contains only 24 images, reliable FID and KID scores could not be calculated. Therefore, we only display the remaining metrics. It can be observed that DiffPC maintains outstanding performance on the Kodak dataset as well. Furthermore, for a more comprehensive validation, we evaluated the statistical fidelity of various methods on the COCO 30K dataset following the settings in (Hoogeboom et al., 2023), as shown in Figure 18. Although the

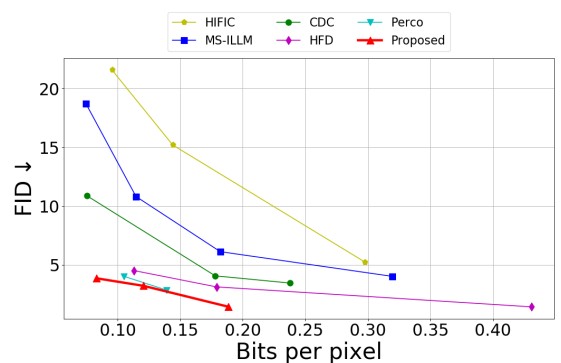

Figure 18: FID tested on COCO 30K.

COCO 30K dataset is not a commonly accepted benchmark for image compression tasks, its capability to assess the statistical fidelity of models has been validated by numerous works, such as

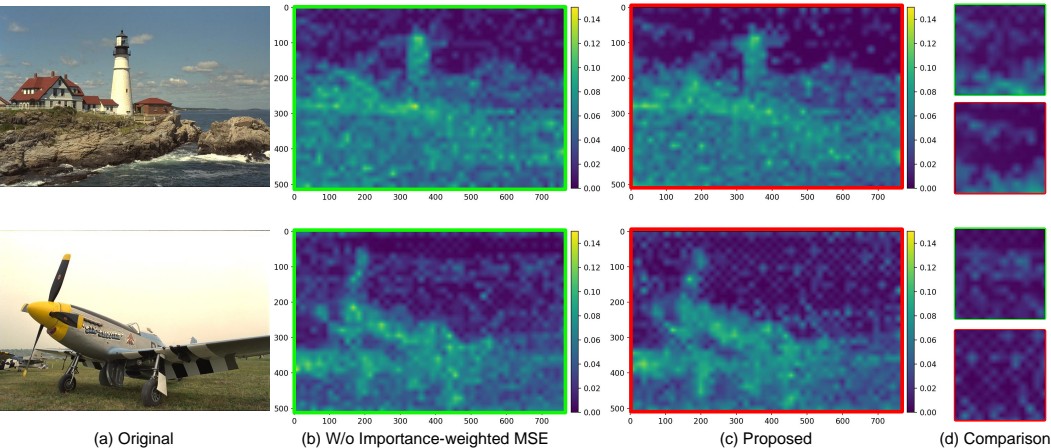

| (a) Original | (b) W/o Importance-weighted MSE | (c) Proposed | (d) Comparison |

Figure 16: The bit rate allocation visualization, where darker colors represent a fewer allocation of bits. To enhance clarity, we interpolated the lower-level features to match the original image dimensions.

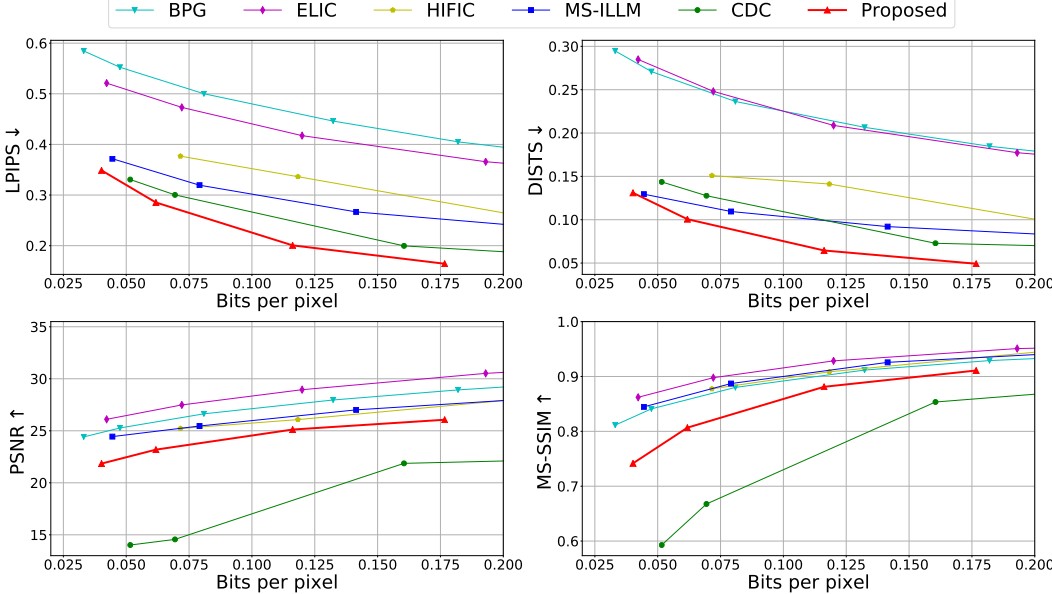

Figure 17: Comparisons of methods across various metrics for the Kodak dataset.

(Reddy et al., 2021). Following the experimental setup in (Hoogeboom et al., 2023; Careil et al., 2024), we partitioned the images into $256 \times 256$ patches. Note that it is challenging to extract meaningful textual descriptions from such small patches, so we used Null Character as a placeholder for textual semantics. Despite this, our model still demonstrated superior statistical fidelity, indicating that DiffPCis not highly dependent on textual semantic information in images.

**Further Qualitative Results:** We showcase additional visual results of DiffPC, HiFiC, and ILLM on three baseline datasets, as illustrated from Figure 19 to Figure 30. Our approach reconstructs details with superior realism using the least number of bits and avoids artifacts that typically arise when reconstructing fine textures.

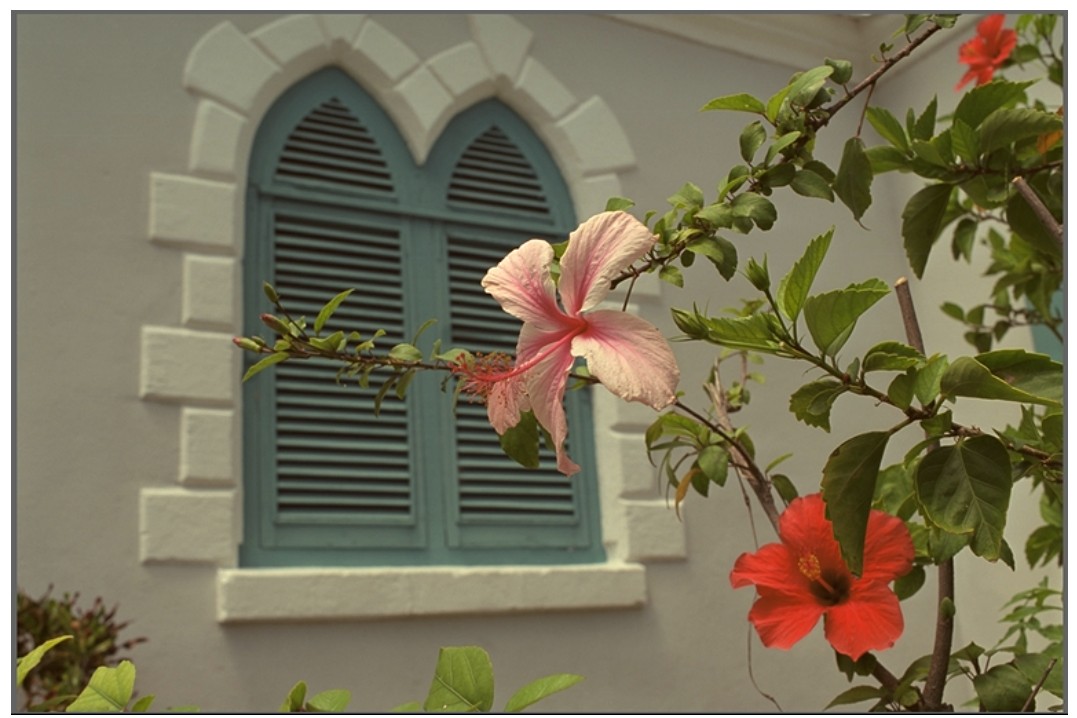

Figure 19: Orignal image (Kodak).

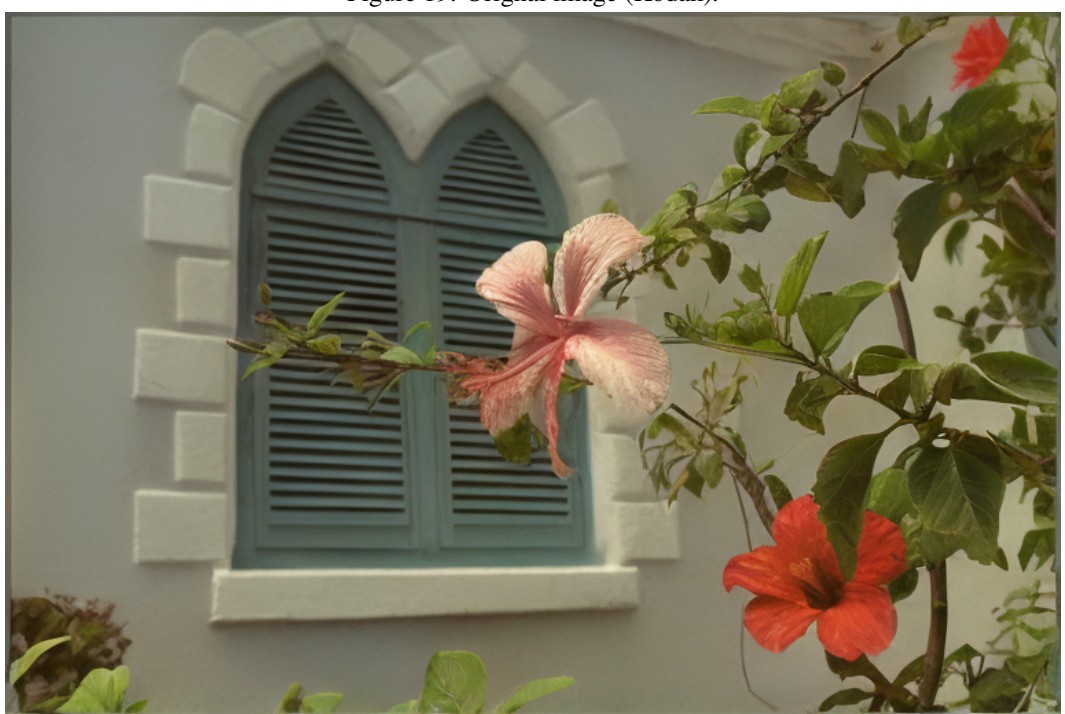

Figure 20: DiffPC(Ours) 0.065 bpp, 0.252 LPIPS

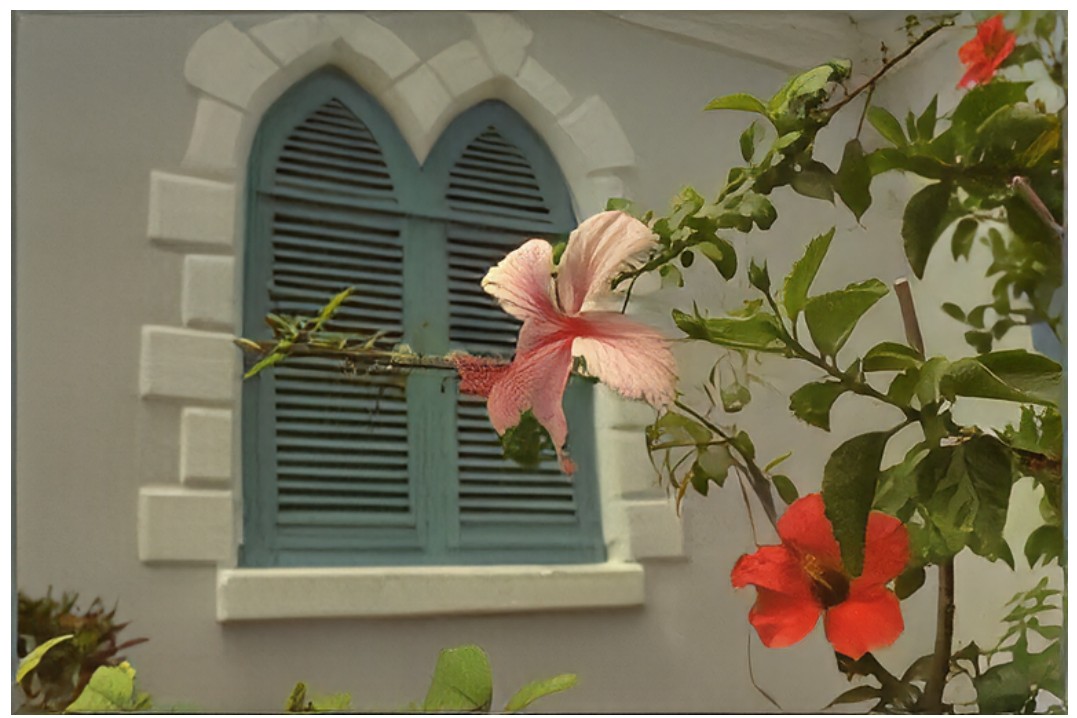

Figure 21: MS-ILLM 0.085 bpp, 0.271 LPIPS

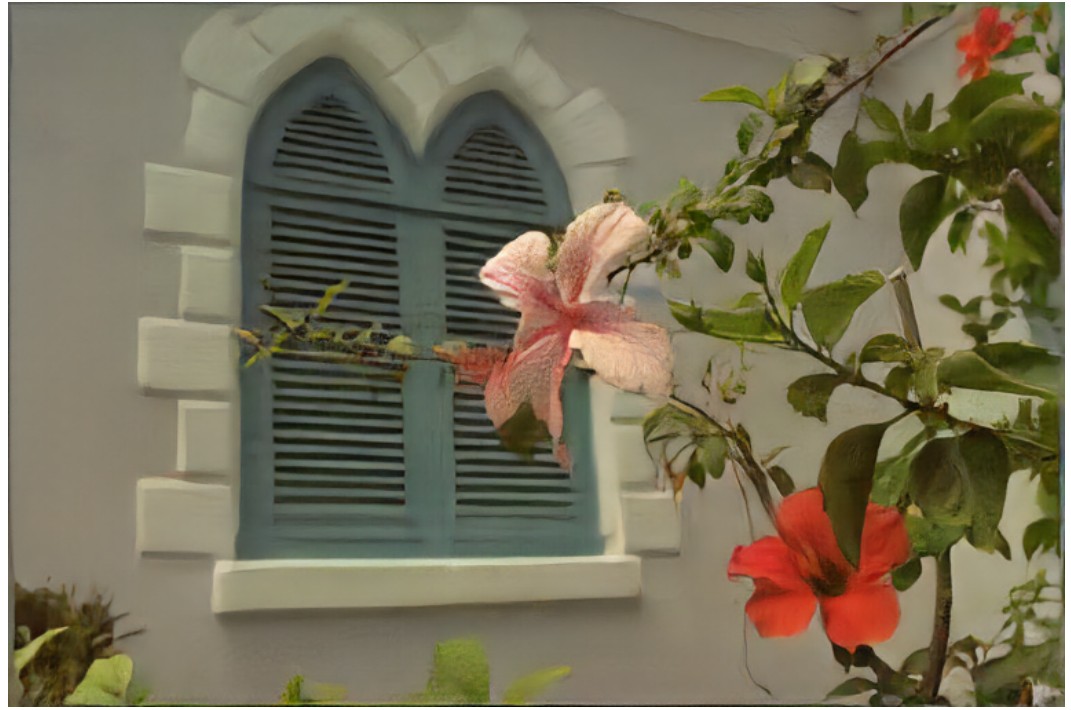

Figure 22: HiFic 0.080 bpp, 0.353 LPIPS

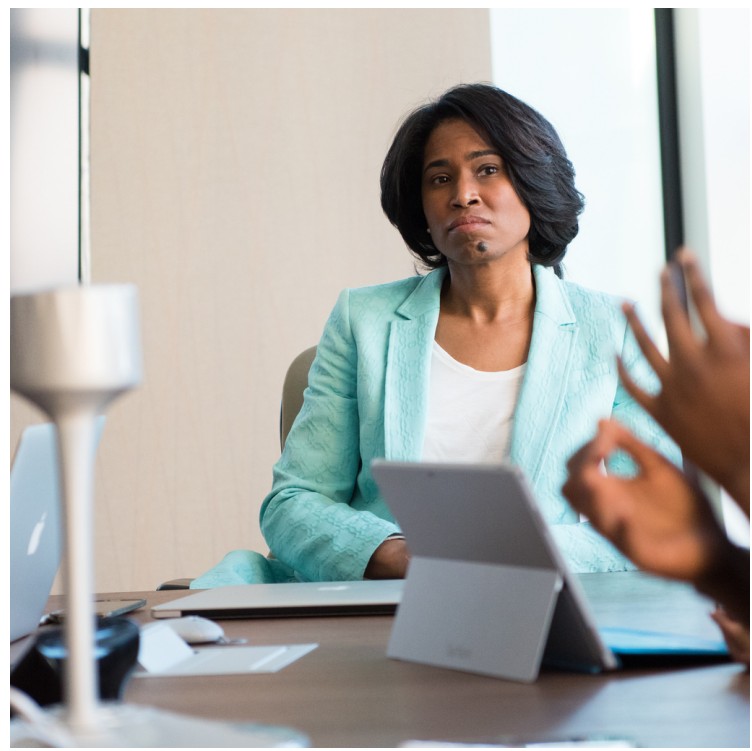

Figure 23: Orignal image (CLIC2020).

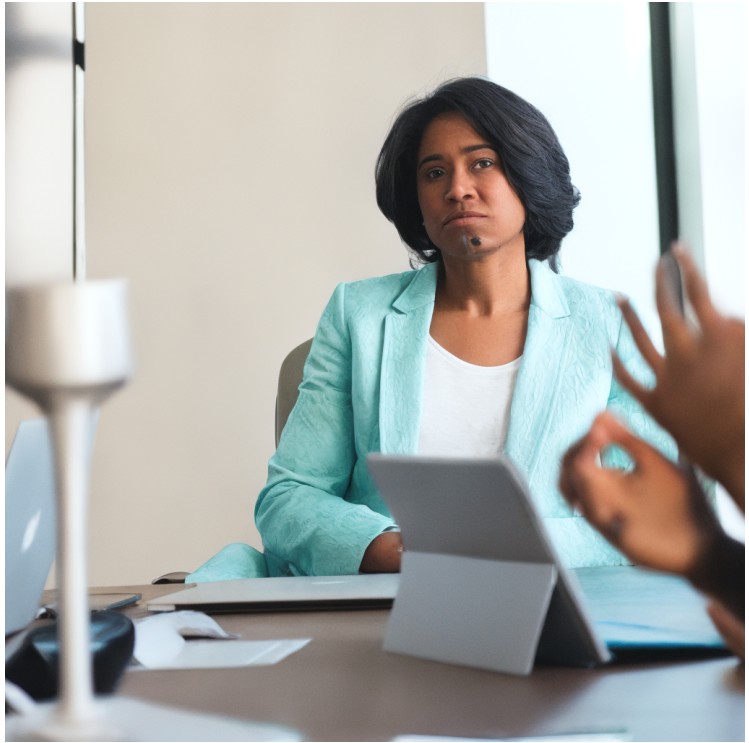

Figure 24: DiffPC(Ours) 0.039 bpp, 0.249 LPIPS

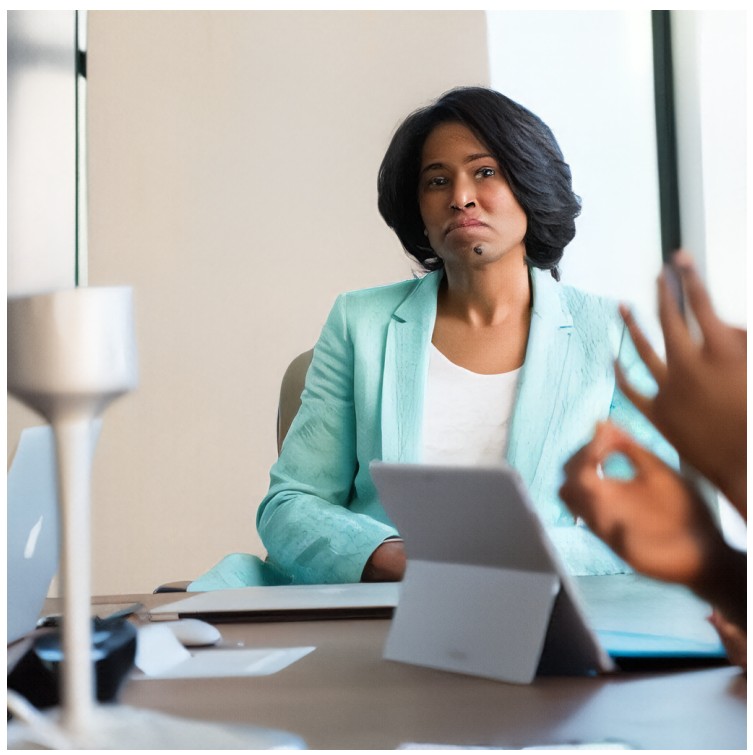

Figure 25: MS-ILLM 0.037 bpp, 0.263 LPIPS

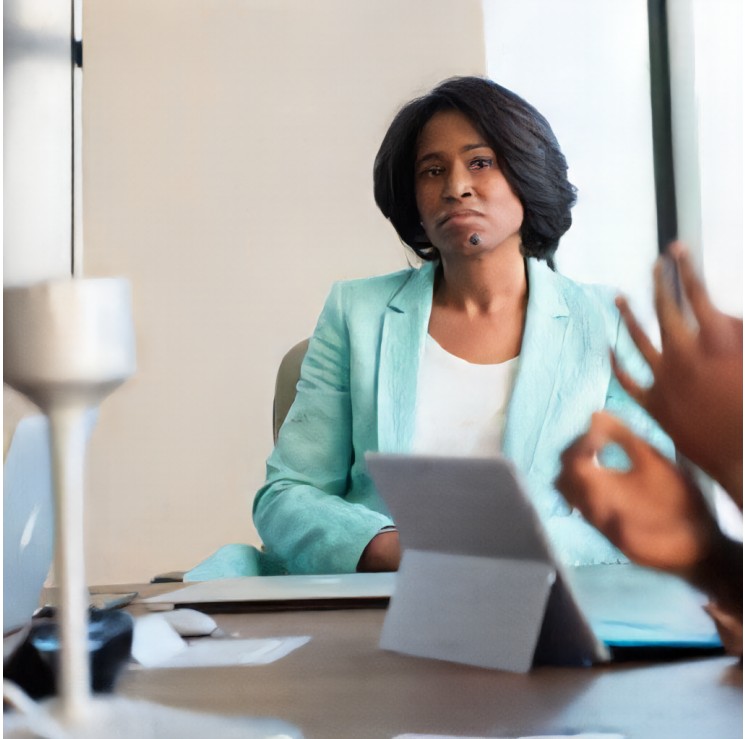

Figure 26: HiFic 0.037 bpp, 0.363 LPIPS

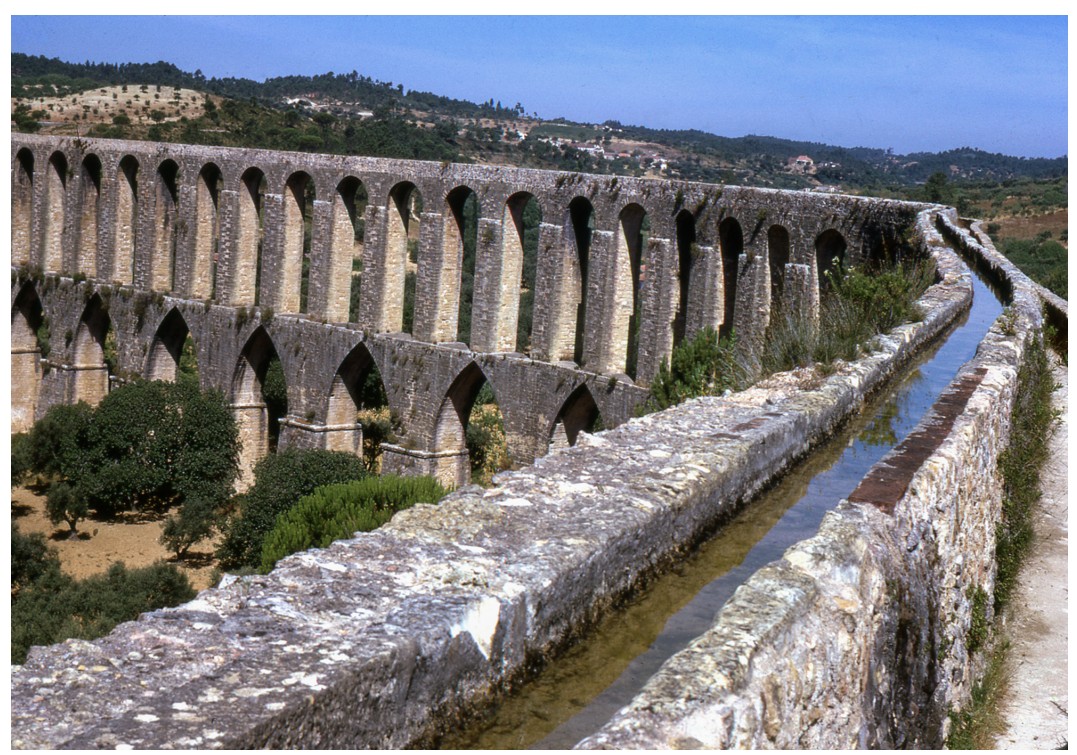

Figure 27: Orignal image (DIV2K).

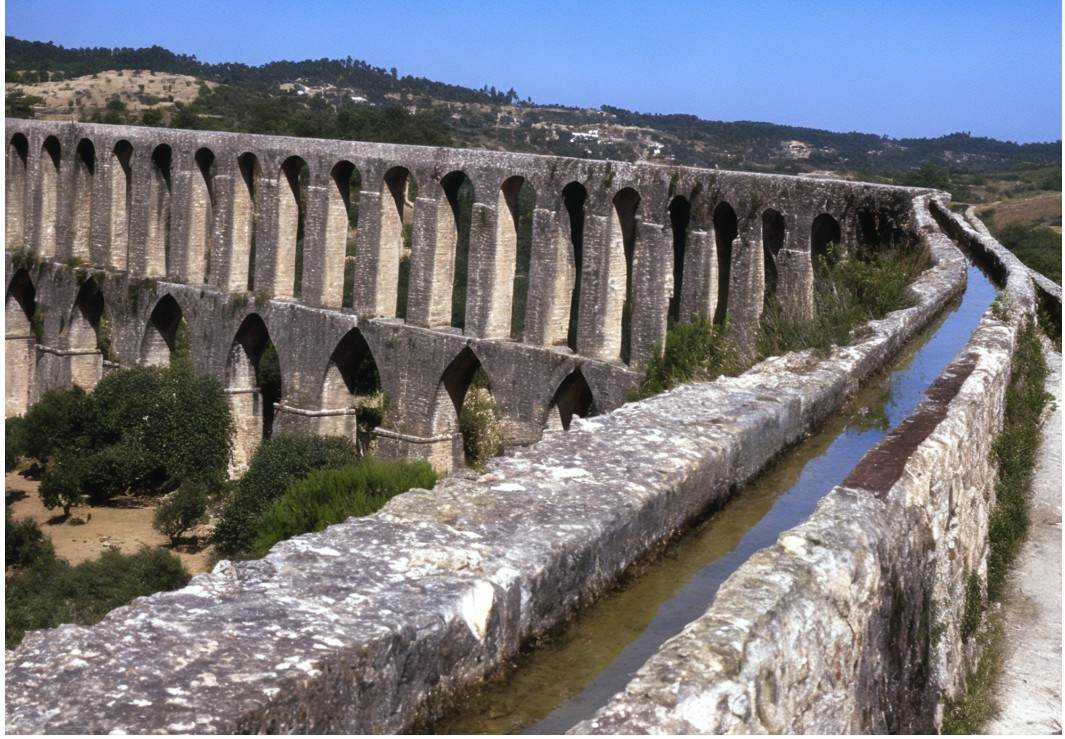

Figure 28: DiffPC(Ours) 0.079 bpp, 0.368 LPIPS

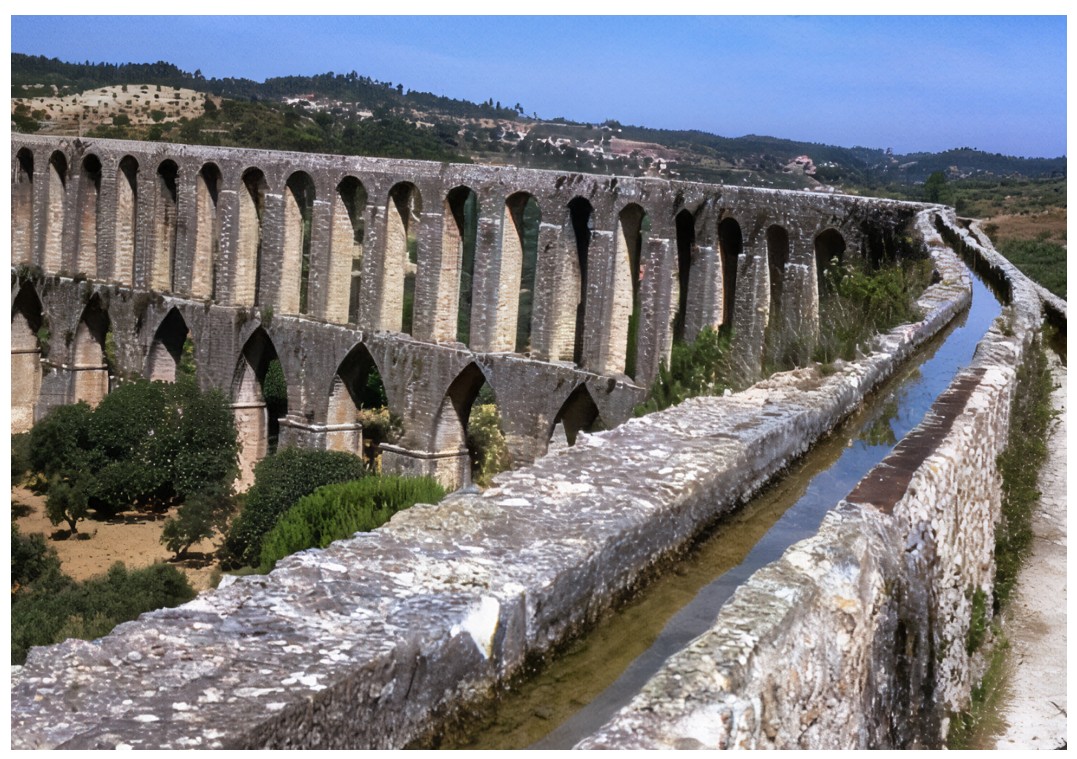

Figure 29: MS-ILLM 0.104 bpp, 0.369 LPIPS

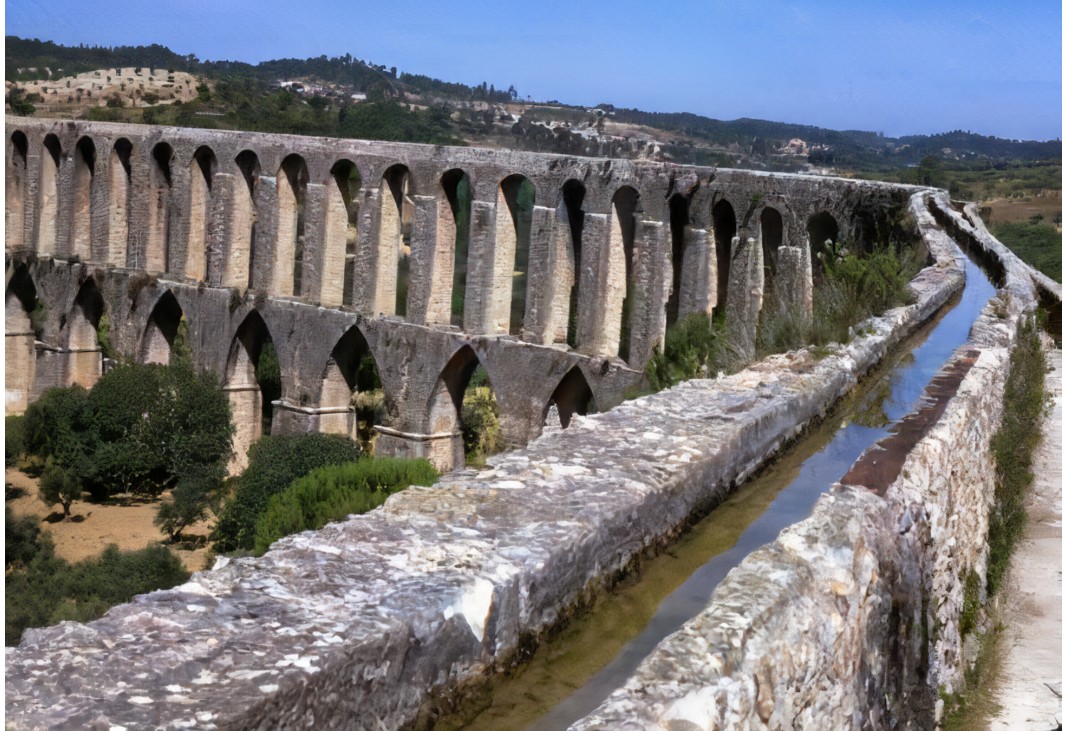

Figure 30: HiFic 0.101 bpp, 0.410 LPIPS

