# OpenReview forum: "DiffPC: Diffusion-based High Perceptual Fidelity Image Compression with Semantic Refinement"
_ICLR.cc/2025/Conference — ICLR 2025 Poster_

### Official Review · Reviewer_suRn · 2024-10-28

**Soundness:** 2
**Presentation:** 2
**Contribution:** 2
**Rating:** 5
**Confidence:** 4

**Summary:**

DiffPC is a novel two-stage image compression framework leveraging stable diffusion models to enhance perceptual fidelity at low bitrates. The model first compresses low-level image details and then integrates semantic features to reconstruct high-quality images. Using modules like a multi-feature compressor, IC-ControlNet, and a semantic pre-embedding module, DiffPC manages to maintain structural and textural consistency.

**Strengths:**

1. State-of-the-Art performance. Demonstrates superior perceptual and statistical fidelity at low bitrates compared to other image compression methods.
2. Efficient semantic integration. The pre-embedding module effectively combines textual and visual semantics, validating that optimizing semantic information has the potential to improve image reconstruction quality. This is a fascinating discovery.

**Weaknesses:**

1. It is recommended that the authors improve the clarity and accuracy of the equations to facilitate reader comprehension. In the "related work" section, some formulas are not precise. For instance, in Eq. 2 and Eq. 3, the variances for the forward and reverse processes are both represented by ${\beta}t$, which is incorrect. Notation such as $DN{\theta}$ is typically represented as $\epsilon_{\theta}$ in existing literature. If the authors are referencing prior work, it is advisable to adhere to standard conventions to improve readability. Additionally, $\textbf{c}$ generally denotes conditional controls, i.e., “text prompts”, which corresponds to $\textbf{s}$ for "textual descriptions" in Eq. 5.

2. For lines 78-87, using `\citep` allows multiple references to be embedded smoothly within a sentence without disrupting the structure. This improves readability, especially when citing multiple sources.

3. Similar works that fine-tune pre-trained diffusion models for low-quality image enhancement include:

   - [1] Lin, X., He, J., Chen, Z., Lyu, Z., Dai, B., Yu, F., ... & Dong, C. (2023) in "DiffBIR: Towards Blind Image Restoration with Generative Diffusion Prior".
   - [2] Wang, Y., Yu, Y., Yang, W., Guo, L., Chau, L. P., Kot, A. C., & Wen, B. (2023) in "ExposureDiffusion: Learning to Expose for Low-Light Image Enhancement" (ICCV).
   - [3] Ma, J., Zhu, Y., You, C., & Wang, B. (2023) in "Pre-trained Diffusion Models for Plug-and-Play Medical Image Enhancement" (MICCAI).

Could the authors further clarify how this manuscript differs from these related works?

**Questions:**

1. In Line 45, are *realism* and *perceptual fidelity* truly equivalent concepts, or could their distinctions be further explored?

2. The experimental section demonstrates that PSNR and SSIM metrics for generative models are not optimal. Could some insightful explanations be provided for why this is the case?

3. Lines 48-49: Are human perception and image semantic consistency necessarily aligned? Not all visual tasks require precise details; for instance, in certain scene understanding tasks, a lower bitrate may be sufficient to convey overall semantics without losing information essential to human perception.

4. The mentions of decoding collapse and condition leakage in the abstract are insightful explanations. Is there corresponding data analysis in the experimental section to support these claims?

---

> ### Author Response · Authors · 2024-11-21
> **Response to Reviewer suRn**
>
> Thank you for your patient reading and feedback! Your recognition of the comprehensiveness of our experimental work and the novelty of our methods is truly motivating for us. We will strive to provide additional explanations of our approach to address any concerns you may have.
>
> ---
>
> **W1-2**:Unclear definition, wrong format
>
> **R1**: Thank you for pointing that out. Upon review, we have identified the issues you mentioned in the manuscript. We have made corrections according to standard definitions to enhance readability. Regarding the definitions of $c$ and $s$, as they have already been explained in Sec 2 and Sec 3.1 for consistency throughout the document, we have decided to maintain these identifiers unchanged. We appreciate your understanding
>
> ---
>
> **W3**:Similar works that fine-tune pre-trained diffusion models for low-quality image enhancement include.
>
> **R2**: Thank you for your feedback! The tasks investigated in the works referenced by [1] [2] [3] all focus on image inverse problems, which differ from the compression task addressed in our study. The commonality lies in the utilization of pre-trained diffusion models as priors.
>
> Specifically, image inverse problems aim to recover the original image by fitting a degradation process under known conditions, whereas image compression tasks involve exploring more compact representations under known prior types to achieve an optimal rate-distortion trade-off, which entails complex bit allocation issues.
>
> Furthermore, our proposed framework incorporates the IC-Control Net and the hybrid semantic refinement module, both of which are innovative structures. These unique components differentiate our work from those referenced articles, as they do not feature similar or identical structures.
>
> ---
>
> **Q1**:In Line 45, are realism and perceptual fidelity truly equivalent concepts, or could their distinctions be further explored?
>
> **R3**: Thank you for your feedback! We have referenced several previous studies [5] [6] and maintained this terminology. Actually, realism does not have a precise definition; we use the term "realism" to refer to images that achieve higher human ratings. Additionally, perceptual metrics such as LPIPS and DISTS significantly enhance human perceptual evaluations and serve as crucial tools for quantitatively assessing photorealism.
> Overall, we believe that equating realism with perceptual quality does not lead to confusion or unfairness in experimental results.
>
> ---
>
> **Q2**:The experimental section demonstrates that PSNR and SSIM metrics for generative models are not optimal. Could some insightful explanations be provided for why this is the case?
>
>
> **R4**: Thank you for your feedback! In Section 1, we mentioned: [4] highlighted the triple trade-off between distortion (PSNR), perceptual realism (perceptual metrics), and bit rate (bpp) at extremely low bitrates. [5] pointed out that at ultra-low bitrates, the significance of distortion (PSNR) as an evaluation metric diminishes, with higher PSNR values not necessarily indicating better image quality [6]. In our experiments in Section 4.2, we demonstrated the best perceptual quality and outperformed existing diffusion baselines regarding distortion metrics like PSNR.
>
> In conclusion, our proposed approach achieves the optimal triple trade-off between perceptual realism, distortion, and bit rate.

---

> > ### Author Response · Authors · 2024-11-21
> > **Author Response (Part II)**
> >
> > **Q3**:Lines 48-49: Are human perception and image semantic consistency necessarily aligned? Not all visual tasks require precise details; for instance, in certain scene understanding tasks, a lower bitrate may be sufficient to convey overall semantics without losing information essential to human perception.
> >
> > **R5**: Thank you for your insightful feedback! We have noted that this sentence may have been ambiguous. What we intended to convey is the human observer's understanding of the semantics in images, which is based on human perception of images. For machine vision tasks, the semantics extracted by neural networks indeed cannot be equated with human perception. To reduce this ambiguity, we will modify the original phrase '......observer’s image semantic consistency' to '......human observer’s image semantic consistency'.
> >
> > ---
> >
> > **Q4**:The mentions of decoding collapse and condition leakage in the abstract are insightful explanations. Is there corresponding data analysis in the experimental section to support these claims?
> >
> > **R6**: Thank you for your feedback. You mentioned our designed TAD module. The ablation experiments for this module were demonstrated in Section 4.3 to showcase both qualitative and quantitative experimental results. From the experimental outcomes, it is evident that the TAD module indeed enhanced the model's performance, particularly in averting image structural degradation arising from conditional leakage, especially at low bit rates. If you seek a more in-depth understanding of the experimental results, please refer to our manuscript. Should you desire additional experiments from us, please inform us promptly, and we shall endeavor to address your inquiries to the best of our ability.
> >
> > ---
> >
> > [1] Lin, X., He, J., Chen, Z., Lyu, Z., Dai, B., Yu, F., ... & Dong, C. (2023) in "DiffBIR: Towards Blind Image Restoration with Generative Diffusion Prior".
> >
> > [2] Wang, Y., Yu, Y., Yang, W., Guo, L., Chau, L. P., Kot, A. C., & Wen, B. (2023) in "ExposureDiffusion: Learning to Expose for Low-Light Image Enhancement" (ICCV).
> >
> > [3] Ma, J., Zhu, Y., You, C., & Wang, B. (2023) in "Pre-trained Diffusion Models for Plug-and-Play Medical Image Enhancement" (MICCAI).
> >
> > [4] Yochai Blau and Tomer Michaeli. Rethinking lossy compression: The rate-distortion-perception
> >
> > [5] Bordin, Tom, and Thomas Maugey. "Semantic based generative compression of images for extremely low bitrates." 2023 IEEE 25th International Workshop on Multimedia Signal Processing (MMSP)
> >
> > [6]  Careil, Marlene, et al. "Towards image compression with perfect realism at ultra-low bitrates." The Twelfth International Conference on Learning Representations. 2024

---

> ### Author Response · Authors · 2024-11-24
> **A Gentle Reminder of the Final Feedback**
>
> Dear Reviewr suRn
>
> Thank you once again for dedicating your valuable time to reviewing our paper and providing constructive comments! We humbly request your brief attention to our responses addressing your concerns. Any feedback you can provide would be immensely appreciated. Moreover, if your inquiries have been adequately addressed, we would be thankful if you could kindly consider adjusting your rating accordingly. Additionally, we remain at your disposal to address any further questions before the conclusion of the rebuttal period.
>
> Sincerely,
>
> The Authors

---

> ### Author Response · Authors · 2024-11-27
> **A Second Reminder of the Post-rebuttal Feedback**
>
> We highly value your initial feedback and understand that you may have a busy schedule. However, we kindly urge you to take a moment to review our responses to your concerns.
>
> In the appendix  A.7.3, we provide additional visual results on bit allocation, along with supplementary ablation studies, illustrating the contribution of our proposed multi-scale compressor to optimizing rate allocation. Our approach utilizes a pre-embedding module to decouple image semantics from quantization codewords, further enhancing the global stability of the diffusion prior by incorporating textual semantics. To the best of our knowledge, this has not been attempted in previous works leveraging diffusion priors.
>
> In conclusion, we humbly request your response once again, as we are eager to engage in further discussion with you.

---

> ### Author Response · Authors · 2024-11-29
> **The Third Warm Reminder of the Post-rebuttal Feedback**
>
> Dear Reviewer suRn,
>
> We notice that all other reviewers have posted their post-rebuttal comments to our response but we still have not received any further information from you. We greatly appreciate your initial comments. We fully understand that you may be extremely busy at this time. However, we kindly request that you take a moment to review our responses addressing your concerns. Due to the passing of the deadline for manuscript revisions, we may not be able to supplement a substantial amount of experiments and visual results to address your concerns, for which we express deep regret. You may peruse our discussions with other reviewers, as this might aid in bolstering your confidence in our work.
>
> In conclusion, we eagerly anticipate your response and are more than willing to engage in further discourse with you.
>
> Best Regards,
>
> Paper 409 Authors

---

> > ### Comment · Reviewer_suRn · 2024-12-01
> >
> > I believe the paper provides valuable empirical insights that could be useful in the field. Regarding the novelty, I will maintain my original score. Good luck with your work!

---

> ### Author Response · Authors · 2024-12-02
> **Thank You for Your Feedback**
>
> Thank you for your response! We have noted your skepticism regarding the novelty of our work. Allow us to reiterate our contributions and clarify the significant distinctions between our research and the works you referenced [1] [2] [3].
>
> We have introduced a novel compression framework featuring a multi-scale compressor that **adjusts rate allocation based on the prior of a pre-trained T-I (Text to Image) LDM**. This approach effectively **reduces the bit rate while preserving high-frequency details, enabling more realistic reconstructions at the decoding stage**. Further, our innovative use of a **pre-embedding module decouples high-level semantics from image prompts**, and combined with textual prompts, it generates a hybrid semantics, **fully leveraging the generative potential of the T-I conditional latent diffusion model**.
>
> Compared to works [1], [2], and [3], our research not only differs in the tasks undertaken but also exhibits significant distinctions in the technical paths and details involved in utilizing diffusion models:
>
> - [1] **focuses on enhancing medical images through a diffusion model that employs the DDPM architecture in the image domain. This work addresses a super-resolution task requiring a known degradation matrix $H$**. Furthermore, it does not leverage the prior knowledge of pre-trained models, opting for full training on the target dataset. Additionally, this study does not utilize any conditional prompts.
> - [2] **explores low-light image enhancement using the DDPM model, proposing a novel analytical sampling process and training objectives based on a known exposure process**. Since our work does not stem from such tasks and lacks known degradation priors to underpin the training of the diffusion model, our training objectives, sampling processes, and the diffusion model backbone employed are markedly different.
> - [3] **involves image enhancement using a pre-trained LDM model, which first learns the degradation process through a restoration network and subsequently samples from the pre-trained LDM's prior to further enhancing the restored images**. In contrast, our work targets compression tasks, where the degradation process is complex and cannot be simulated through a simple network as in super-resolution tasks. Moreover, [3] does not incorporate any high-level semantic prompts, which we believe inevitably leads to suboptimal results.
>
> **In summary, there are clear differences in applying diffusion frameworks across various tasks. We believe it is unreasonable to diminish the novelty of our work solely due to the shared use of a diffusion model backbone; however, we deeply respect your viewpoints and comments.** We would be most grateful if you are willing to engage in further discussion.
>
> ___
>
> [1] Ma, J., Zhu, Y., You, C., & Wang, B. (2023) in "Pre-trained Diffusion Models for Plug-and-Play Medical Image Enhancement" (MICCAI).
>
> [2] Wang, Y., Yu, Y., Yang, W., Guo, L., Chau, L. P., Kot, A. C., & Wen, B. (2023) in "ExposureDiffusion: Learning to Expose for Low-Light Image Enhancement" (ICCV).
>
> [3]  Lin, X., He, J., Chen, Z., Lyu, Z., Dai, B., Yu, F., ... & Dong, C. (2023) in "DiffBIR: Towards Blind Image Restoration with Generative Diffusion Prior".

---

### Official Review · Reviewer_1ruM · 2024-10-28

**Soundness:** 3
**Presentation:** 2
**Contribution:** 2
**Rating:** 6
**Confidence:** 5

**Summary:**

(1).This paper proposes a diffusion-based high-perceptual fidelity image compression with semantic refinement.
(2).The proposed method uses the low-level image information to ensure fidelity reconstruction,  and uses high-level texture semantics and robust prior in pre-trained latent stable diffusion to ensure realism reconstruction.
(3).Extensive experiments demonstrate its effectiveness in reconstructing images with high perceptual quality texture at extremely low bitrates

**Strengths:**

Originality: This paper explores the pre-trained latent diffusion model using semantic control signals for extreme image compression. This method clearly explains that two types of semantics, namely low-level image information (visual semantics) and high-level semantics (textual semantics), can support image compression.

Quality: The method is well designed and integrates the pre-trained latent diffusion model and semantic information for extreme image compression.

Clarity: The paper is generally well written and organized.

Significance: The significance of this work is profound as it addresses the extreme image compression using pre-trained latent diffusion models and semantic information.

**Weaknesses:**

(1). The authors do not describe some representative perceptual image compression methods corresponding to their work in the related work.
[1]. Multi-modality deep network for extreme learned image compression. AAAI 2023.
[2]. Extreme Image Compression using fine-tuned VQGANs. Data Compression Conference 2024.
[3]. Towards Extreme Image Compression with Latent Feature Guidance and Diffusion Prior. IEEE TCSVT 2024.
[4]. Consistency Guided Diffusion Model with Neural Syntax for Perceptual Image Compression. ACMMM 2024.
[5]. HybridFlow: Infusing Continuity into Masked Codebook for Extreme Low-Bitrate Image Compression. ACMMM 2024.
[6]. Neural Image Compression with Text-guided Encoding for both Pixel-level and Perceptual Fidelity. ICML 2024.

(2). In Figure 2, it is confusing that you use the same color of block to represent cross-attention layers. The pre-embedding module is not shown in Figure 2. The ICCN (i.e., IC-ControlNet) is not explained in Figure 2 or your paper.

(3). In Eq.(7), the c ̂_x is a degraded image instead of features, which is the input of an image encoder, as shown in Figure 2. You need to add the image encoder to your equation.

(4). In the part ``IC-ControlNet and Time-aware Decoupling” of section 3.2, the conditions are integrated with noise and processed through residual blocks before entering the main network. I cannot find any residual blocks as you say in Figure 9.

(5). In Section 3.3, the symbol 〖text〗_x should be shown in Figure 2. The semantic pre-embedding module should be marked in Figure 2 (the same as (2)). The blended semantic s_x should be added to Figure 2 for clarity.

(6). Please compare your method with VQGAN-based compression (DCC2024), TACO (ICML2024), and DiffEIC (IEEE TCSVT2024) and add them to Figures 4 and 5. VQGAN-based IC performs extreme image compression using codebook prior. TACO uses the text prior and DiffEIC uses the diffusion prior, which are very relevant to your work.

Reference papers:
[1]. Extreme Image Compression using fine-tuned VQGANs. Data Compression Conference 2024.
[2]. Towards Extreme Image Compression with Latent Feature Guidance and Diffusion Prior. IEEE TCSVT 2024.
[3].Neural Image Compression with Text-guided Encoding for both Pixel-level and Perceptual Fidelity. ICML 2024.

(7). What is TAC in your ablation study?  I think it is the TAD module in your procedure.

(8). The text can assist image compression. I think you need to further analyze the text description for image compression. For example, you can use different caption methods to generate captions, which are used for your compression framework (test the text robustness of your method).

(9). The author should compare the proposed method with state-of-the-art methods in terms of model complexity.

**Questions:**

(1). Why do you need intermediate features f1 and f2 for a multi-feature compressor? Maybe using a simple compressor without multi-scale features is simple but effective. Please add an ablation study to validate its effectiveness for image compression.

(2). Could you give a visual comparison of using importance-weighted MSE and standard MSE? It ensures proper bit allocation as you describe in your paper. You can visualize the bit allocation map to explain this.

---

> ### Author Response · Authors · 2024-11-21
> **Response to Reviewer 1ruM**
>
> We greatly appreciate your constructive suggestions and recognition of the innovativeness of our work! Your acknowledgment of the significance and comprehensiveness of our experiments is truly inspiring! Your feedback contributes to the refinement of our endeavors, and we will exert utmost diligence in addressing your inquiries and dispelling any uncertainties.
>
> ---
>
> **W1**: The authors do not describe some representative perceptual image compression methods corresponding to their
> work in the related work.
>
> **R1**: Thank you for your suggestions. We have thoroughly considered these relevant works and incorporated them into our related work section.
>
> ---
> **W2,3,4,5,7**: typo error &  issue of article clarity
>
> **R2**: Thank you for your careful reading and feedback! We apologize for any reading obstacles caused by typo errors. Following your advice, we have included citations to relevant works in the revised version, corrected the corresponding descriptions. This includes changes to Fig. 2 and Fig. 9, as well as adjusting all references from TAC to TAD.
>
> ---
> **W6**:Please compare your method with VQGAN-based compression (DCC2024), TACO (ICML2024), and DiffEIC (IEEE TCSVT2024) and add them to Figures 4 and 5. VQGAN-based IC performs extreme image compression using codebook prior. TACO uses the text prior and DiffEIC uses the diffusion prior, which are very relevant to your work.
>
> **R3**: Your suggestions are quite meaningful and assist us in further refining our experiments. Following your advice, we have incorporated comparisons with these latest baselines in Figures 4 and 5. Here, we present some of the results:
>
> **Table 1.** Tests performed on CLIC20. The metric is BD-rate（%）, based on DIffPC. A higher BD-rate indicates a larger gap with DiffPC.
> |    Model     |  LPIPS  | DISTS  |   FID   |   KID   |  PSNR   | MS-SSIM |   Total   |
> | :----------: | :-----: | :----: | :-----: | :-----: | :-----: | :-----: | :-------: |
> |  DiffEIC[1]  | 0.65645 | 2.3841 | 10.4487 | 117.297 | 42.108  | 0.9902  | 173.88445 |
> |   TACO[2]    | 83.4615 | 80.241 | 62.955  | 370.724 | -40.745 | -30.434 | 526.2025  |
> | Ours(DiffPC) |    0    |   0    |    0    |    0    |    0    |    0    |     0     |
>
> We conducted our experiments using the official provided code and checkpoints. Observations on the CLIC20 dataset reveal that our proposed DiffPC consistently outperforms the baseline schemes in perceptual metrics. While DiffPC shows slightly lower performance in pixel-level distortion metrics (PSNR) compared to TACO, it exhibits significant advantages in statistical fidelity and perceptual fidelity.
>
>
> We have plotted the test results of [3] in Figures 4 and 5. Due to the relatively aggressive fine-tuning approach employed by the VQGAN based [3], the available bitrate range is limited, making it challenging to select sufficient data points for reliable BD-rate computation. Furthermore, we have noticed that if the bitrate approaches or exceeds the maximum bitrate range accommodated by [3], the performance of VQGAN may even deteriorate as the bitrate increases, indicating concerns regarding its bitrate generalization performance. In contrast, DiffPC demonstrates stronger bitrate generalization capabilities and achieves comparable ultra-low bitrate performance to VQ-GAN based on less training data (0.08Million vs. 1.2Million).
>
> ---
>
> [1]Towards Extreme Image Compression with Latent Feature Guidance and Diffusion Prior. IEEE TCSVT 2024.
>
> [2]Neural Image Compression with Text-guided Encoding for both Pixel-level and Perceptual Fidelity. ICML 2024
>
> [3]Extreme Image Compression using fine-tuned VQGANs. Data Compression Conference 2024

---

> > ### Author Response · Authors · 2024-11-21
> > **Author Response (Part II)**
> >
> > **W8**:The text can assist image compression. I think you need to further analyze the text description for image compression. For example, you can use different caption methods to generate captions, which are used for your compression framework (test the text robustness of your method).
> >
> > **R4**: Thank you for your insightful commentary! In Section A.7.2, we conducted supplementary experiments and explanations on the robustness of textual prompts. Next, we proceed with some summary discussions.
> >
> > We have two primary reasons for employing textual semantics. Firstly, text can offer global semantic information, maintaining the stability of generated samples. Secondly, through the pre-embedding module, we can reduce the distance between image and text semantics in the embedding space, allowing image semantics to better modulate the denoising model.
> >  Given that our semantic control flow involves a blend of images and text, our approach exhibits considerable robustness to diverse textual inputs:
> >
> >
> > **Table 2.** Text robustness evaluation. Tested on KodaK with a BD-rate (%) based on DiffPC(Ours).
> > |                    Text type                    | LPIPS  | DISTS  |  PSNR  |
> > | :---------------------------------------------: | :----: | :----: | :----: |
> > |       BLIP2 (5$\leq$words count $\leq$ 10)       | 1.686  | 1.5792 | 0.8911 |
> > |                    OFA-tiny                     | 1.742  | -0.311 | 0.320  |
> > |                  Default text                   | 4.5024 | 5.4548 | 0.4472 |
> > |                  Random string                  | 19.577 | 35.816 | 82.989 |
> > | BLIP2 (10$\leq$words count$\leq$ 30) （DiffPC） |   0    |   0    |   0    |
> >
> > The table above represents our quantitative experiments on the robustness of text prompt. We employed four different textual descriptions: *（1）BLIP2 (5 < words count < 10)*, where the output word count of BLIP2 is forcibly shortened to within 10 words; *（2）OFA-tiny*, utilizing [1] as a new image captioning model to obtain textual prompt for the images; *（3）Default text*, where all images are uniformly described with the default text "high quality, extreme detail"; *（4）Random string*, using random strings of equal length (20 characters).
> >
> > It is evident that as long as the textual descriptions align with the semantics of the images, they generally do not diminish the model's reconstruction performance. Even with the use of default textual propmts, the model can still benefit from the text. This aligns with our expectations: reasonable textual semantics can assist in the embedding of image semantics, enabling image semantics to be more effectively guided by the denoising model through the cross-attention layers. However, when textual semantics are chaotic and disordered, this can disrupt the model's ability to control image semantics, leading to a significant performance decline. To further elucidate the relationship between textual semantics and image semantics, we have conducted additional ablation experiments：
> >
> > **Table 3.** The ablation experiments were conducted on the DIV2K dataset, utilizing the BD-rate (%) metric with DiffPC as the baseline.
> >
> > |      Model       |  LPIPS  | DISTS  |   FID   |   KID   |
> > | :--------------: | :-----: | :----: | :-----: | :-----: |
> > | W/o img semantic | 19.8428 | 47.524 | 30.302  |  60.66  |
> > | W/o txt semantic | 9.7942  | 17.698 | 13.2731 | 34.3854 |
> > |      DiffPC      |    0    |   0    |    0    |    0    |
> >
> > It is evident that both types of semantics contribute to performance improvements, with image semantics providing a slightly greater enhancement. We have included these ablation experiments in the revised experimental section. For more detailed information, please refer to Sec4.3 and Sec A.7.2 of the revised version.
> >
> > ---
> >
> > [1] Wang, Peng, et al. "Ofa: Unifying architectures, tasks, and modalities through a simple sequence-to-sequence learning framework." International conference on machine learning. PMLR, 2022.
> >
> > ---
> >
> > **W9**:The author should compare the proposed method with state-of-the-art methods in terms of model complexity
> >
> > **R5**: Thank you for your constructive suggestions! To alleviate your concerns, we have included comparative experiments on model complexity with state-of-the-art solutions in the "Joint Response about Model Complexity." These experiments have also been added to Appendix Sec A.5. For a more detailed discussion, you can refer to our revised version.

---

> > > ### Author Response · Authors · 2024-11-21
> > > **Author Response (Part III)**
> > >
> > > **Q1**:Why do you need intermediate features f1 and f2 for a multi-feature compressor? Maybe using a simple compressor without multi-scale features is simple but effective. Please add an ablation study to validate its effectiveness for image compression.
> > >
> > > **R6**: We appreciate your insightful feedback. In response to your concerns, we will initially conduct quantitative analyses by means of ablation experiments on the multi-scale feature structure:
> > >
> > > **Table 4.** The ablation experiments were conducted on the DIV2K dataset, utilizing the BD-rate (%) metric with DiffPC as the baseline.
> > > |      Model       | LPIPS | DISTS |  FID   |   KID    |   Paramters   |
> > > | :--------------: | :---: | :---: | :----: | :------: | :-----------: |
> > > | W/o Muti-feature | 5.455 | 7.286 | 6.9316 | 11.24091 | 24.53M (13%↓) |
> > > |  Ours (DiffPC)   |   0   |   0   |   0    |    0     |    28.20M     |
> > >
> > > The impact of removing the Multi-feature component on the model's performance is apparent. Although this reduction may result in a decrease in the model's parameter count, we consider it advantageous to introduce a modest number of parameters to ensure a consistent improvement in performance. Moreover, we have incorporated the results of the ablation experiments into Figure 8 in the main text and provided a detailed analysis of this module.
> > >
> > > Following this, we will elucidate our viewpoint on this multi-scale structure.
> > >
> > > - The incorporation of multi-scale features, as evidenced in several prior studies [1][2], has been confirmed for its efficacy in expanding the receptive field and retaining high-frequency details. Our objective is to extend the perceptual scope of the encoding phase towards the original image through the integration of multi-scale features.
> > > -  Moreover, the initial design principle of Stable diffusion for image generation inherently leads to the creation of high-frequency features in the generated images instead of conserving them from the source image. These high-frequency components are typically lost during the encoding phase by $\mathcal{E}(\cdot)$, whereas our specially devised multi-scale architecture is crafted to fully restore these high-frequency details.
> > > - In low bitrate scenarios, we observed that the constraints on distortion are relatively weak, leading to the significant loss of fine details originally contained in the signal $c$. We aim to complement these high-frequency details through a multi-scale structure, directing the compressor to focus more on textures rather than low-frequency semantics. The low-frequency semantics can be compensated for in subsequent stages through diffusion priors and semantic enhancement in the second phase.
> > >
> > > ---
> > >
> > > [1] Li, Feng, et al. "Lite detr: An interleaved multi-scale encoder for efficient detr." *Proceedings of the IEEE/CVF conference on computer vision and pattern recognition*. 2023.
> > >
> > > [2] Chang, Jia-Ren, and Yong-Sheng Chen. "Pyramid stereo matching network." *Proceedings of the IEEE conference on computer vision and pattern recognition*. 2018.

---

> > > > ### Author Response · Authors · 2024-11-21
> > > > **Author Response (Part IV)**
> > > >
> > > > **Q2**:Could you give a visual comparison of using importance-weighted MSE and standard MSE? It ensures proper bit allocation as you describe in your paper. You can visualize the bit allocation map to explain this.
> > > >
> > > > **R7**: Thank you for your valuable insights and suggestions. In the revised version of the paper, we have provided visualizations of bit allocation in the Sec A.7.3, confirming the effectiveness of the importance-weighted loss proposed. Compared to not using $\mathcal{L}_{imp}$, DiffPC allocates fewer bits in flat regions (such as the sky) and concentrates bits more in areas with complex textures.

---

> ### Author Response · Authors · 2024-11-24
> **A Gentle Reminder of the Final Feedback**
>
> Dear Reviewr 1ruM
>
> Thank you once again for dedicating your valuable time to reviewing our paper and providing constructive comments! As the end of the discussion period approaches, we kindly ask if our responses have satisfactorily addressed your concerns. Your feedback would be greatly appreciated, and we would be delighted to engage in further discussions if needed.
>
> Sincerely,
>
> The Authors

---

> > ### Comment · Reviewer_1ruM · 2024-11-26
> >
> > Thank you for your response and you have addressed all my questions.

---

> > > ### Author Response · Authors · 2024-11-26
> > > **Thanks to reviewer 1ruM**
> > >
> > > We appreciate your response and the contribution your feedback has made to improving our work! If all your queries have been addressed, we kindly ask you to consider raising your rating. If you still have any doubts or reservations about our work, we are more than willing to engage in further discussion with you.

---

> ### Comment · Reviewer_1ruM · 2024-11-27
>
> I think you need to go over your manuscript again and make sure there are no mistakes. I have improved the score and wish you good luck.

---

> > ### Author Response · Authors · 2024-11-27
> > **Thanks to reviewer 1ruM**
> >
> > Thank you very much for your response! We will review the manuscript once more to ensure that no errors are present. We greatly appreciate your comments and contributions to our work!

---

### Official Review · Reviewer_GPEP · 2024-10-31

**Soundness:** 3
**Presentation:** 2
**Contribution:** 3
**Rating:** 6
**Confidence:** 5

**Summary:**

The paper proposes a diffusion-based image compression framework, DiffPC, aimed to achieve high perceptual fidelity at low bit rates. This model leverages both low-level degraded image representation and high-level hybrid semantic representation to control pre-trained Stable Diffusion. Experimental results show that the proposed method surpasses existing methods across various benchmarks, delivering superior perceptual quality and statistical fidelity.

**Strengths:**

1. The usage of high-level hybrid semantic representation in diffusion-based image compression methods is novel. I appreciate the idea of extracting semantic information from $\hat{c}_x$.

2. The paper effectively communicates the purpose and rationale behind each module in the DiffPC framework, providing clear justifications for their functionality.

3. The paper includes thorough experiments with quantitative and qualitative metrics across multiple datasets, validating the effectiveness of DiffPC.

**Weaknesses:**

1. The paper lacks a thorough comparison and analysis of computational complexity. As diffusion models inherently introduce high decoding complexity, it is essential to evaluate and compare this aspect against previous SOTA methods.

2. The paper has some issues related to writing clarity and accuracy of details. Please refer to questions 1-9 for specifics.

3. There are issues with citation formatting in certain parts of the paper. For instance, on page 2, line 079, the sentence ``… generative adversarial frameworks Mentzer et al. (2020); Muckley et al. (2023)`` should be formatted as ``… generative adversarial frameworks (Mentzer et al., 2020; Muckley et al., 2023).``

4. Other existing image compression methods [1, 2] could be discussed and compared.

[1] Zhiyuan Li, Yanhui Zhou, Hao Wei, Chenyang Ge, and Jingwen Jiang. Towards extreme image compression with latent feature guidance and diffusion prior. IEEE Transactions on Circuits and Systems for Video Technology, 2024.

[2] Hagyeong Lee, Minkyu Kim, Jun-Hyuk Kim, Seungeon Kim, Dokwan Oh, and Jaeho Lee. Neural image compression with text-guided encoding for both pixel-level and perceptual fidelity. In International Conference on Machine Learning, 2024

**Questions:**

1. On page 2, line 097, ``Careil et al. (2023) ...`` PerCo was published in ICLR2024.

2. On page 3, line 138, in the sentence ``He et al. (2022), based on the orthogonality of features in channel and spatial dimensions, devised an asymmetric autoregressive entropy encoder..`` there are two periods at the end.

3. On page 4, line 183, there is a grammatical error in the sentence: ``Here, c and s respectively refer to low-level image controls (e.g., image contours, image degradation), with s denoting high-level semantic controls (e.g., textual descriptions of images, category label).``

4. On page 10, line 520, in the sentence ``(2) w/o TAC: Removing the TAC ...``, should `TAC` actually be `TAD`?

5. On page 11, line 545, ``En d-to-end optimized image compression.`` There is an extra space that should be removed.

6. Is Eq. (3) accurate? $\mathcal{DN}_{\theta}(z_t, t)$ is the predicting noise rather than the mean.

7. In Eq. (13), what does $\eta$ denote in $TAD_{\eta}$?

8. In Figure 10(a), it appears that an arrow from the conv3×3 layer to the Spatial Transformer is missing. In Figure 10(b), there are two instances of $\varepsilon$. Should they be represented by different symbols to avoid confusion?

9. On page 4, line 178, the introduction to Stable Diffusion is incomplete, as it does not explain how $\hat{z}_0$ is obtained within Stable Diffusion.

10. The authors mention that ``the importance-weighted loss assigns more bits to texture details, enabling a more precise reconstruction of these features.`` Could the authors provide a visualization result of the bit allocation to support this conclusion?

11. What impact do multi-scale features have on performance? In Eq. (12), the loss $L_{imp}$ is calculated based on the difference between $z_0$ and $c$, where multi-scale features do not appear to be necessary. Additionally, the authors have not demonstrated the role or necessity of multi-scale features in the ablation study. Clarifying these points would strengthen the paper’s argument for including multi-scale features.

12. In Eq. (13), how is the initial value of $w$, specifically $\sigma_{z_0}^2$, determined in practice? Additionally, how do you ensure that the trainable hyperparameter $w$ correctly learns to achieve precise bit allocation?

13. This module adopts a pre-embedding approach that combines textual semantics with visual semantics. Could the authors clarify the individual roles of textual semantics and visual semantics? Additionally, what impact would using only textual semantics or only visual semantics have on performance?

---

> ### Author Response · Authors · 2024-11-21
> **Response to Reviewer GPEP**
>
> We greatly appreciate your constructive suggestions and recognition of the innovation in our work! Your acknowledgment of the innovation and comprehensiveness of our experiments is truly inspiring. The suggestion you made regarding additional ablation experiments has sparked our enthusiasm. We will exert our utmost efforts to address your inquiries and alleviate any concerns you may have.
>
> ---
>
> **W1**: The paper lacks a thorough comparison and analysis of computational complexity. As diffusion models inherently introduce high decoding complexity, it is essential to evaluate and compare this aspect against previous SOTA methods.
>
> **R1**: Thank you for your constructive suggestions! To alleviate your concerns, we have included comparative experiments on model complexity with state-of-the-art solutions in the "Joint Response about Model Complexity." These experiments have also been added to Appendix Sec A.5. For a more detailed discussion, you can refer to our revised version.
>
> ---
>
> **W2**: Other existing image compression methods [1, 2] could be discussed and compared.
>
> **R2**: Thank you very much for your suggestions; they are invaluable in helping us enhance our experiments further. Following your advice, we have incorporated comparisons with these latest baselines. Here, we present some of the results:
>
> **Table 1.** Tests performed on CLIC20. The metric is BD-rate（%）, based on DIffPC. A higher BD-rate indicates a larger gap with DiffPC.
> |    Model     |  LPIPS  | DISTS  |   FID   |   KID   |  PSNR   | MS-SSIM |   Total   |
> | :----------: | :-----: | :----: | :-----: | :-----: | :-----: | :-----: | :-------: |
> |  DiffEIC[1]  | 0.65645 | 2.3841 | 10.4487 | 117.297 | 42.108  | 0.9902  | 173.88445 |
> |   TACO[2]    | 83.4615 | 80.241 | 62.955  | 370.724 | -40.745 | -30.434 | 526.2025  |
> | Ours(DiffPC) |    0    |   0    |    0    |    0    |    0    |    0    |     0     |
>
> We conducted our experiments using the official provided code and checkpoints. It can be observed that on the CLIC20 dataset, our proposed DiffPC consistently outperforms the baseline solutions in perceptual metrics. While DiffPC may lag slightly behind TACO in pixel-level distortion metrics (PSNR), it exhibits significant advantages in statistical fidelity and perceptual fidelity. Furthermore, we have included the above experiments in the revised experimental section, where you can find more results in Sec 4.2.
>
> [1] Zhiyuan Li, Yanhui Zhou, Hao Wei, Chenyang Ge, and Jingwen Jiang. Towards extreme image compression with latent feature guidance and diffusion prior. IEEE Transactions on Circuits and Systems for Video Technology, 2024.
>
> [2] Hagyeong Lee, Minkyu Kim, Jun-Hyuk Kim, Seungeon Kim, Dokwan Oh, and Jaeho Lee. Neural image compression with text-guided encoding for both pixel-level and perceptual fidelity. In International Conference on Machine
> Learning, 2024
>
> ---
>
> **Q1-9**: Typo error & writing clarity.
>
> **R3**: Once again, we express our gratitude for taking the time to review our paper. In the revised version, we have rectified all the errors you mentioned and have clarified some ambiguous statements and images. Next, we will provide specific responses to some of the issues raised.
>
> Q4: ...Removing the TAC ... , should TAC actually be TAD ?
>
> R: This is a typo error; we have now corrected all instances of "TAC" in the images and sections to "TAD."
>
> Q6:  Is Eq. (3) accurate? $\mathcal{D}$ is the predicting noise rather than the mean
>
> R:  Thank you for pointing that out. We have revised the wording and replaced the denoising network $\mathcal{D}(\cdot)$ with the neural network $\mathcal{M}(\cdot)$, providing a more precise explanation.
>
> Q7: In Eq. (13), what does $\eta$  denote in $\mathbf{TAD}_{\eta}$?
>
> R: Here, $\eta$ represents the parameters of the network $\mathbf{TAD}{\eta}$, emphasizing that $\mathbf{TAD}{\eta}$ is a neural network with trainable parameters.
>
> ---
>
> **Q10**: The authors mention that the importance-weighted loss assigns more bits to texture details, enabling a more precise reconstruction of these features. Could the authors provide a visualization result of the bit allocation to support this conclusion?
>
> **R3**: Thank you very much for your insightful suggestions! In the revised version of the paper, we have provided visualizations of bit allocation in the Sec A.7.3, confirming the effectiveness of the importance-weighted loss proposed. Compared to not using $\mathcal{L}_{imp}$ , DiffPC allocates fewer bits in flat regions (such as the sky) and concentrates bits more in areas with complex textures.

---

> ### Author Response · Authors · 2024-11-21
> **Author Response (Part II)**
>
> **Q11**: What impact do multi-scale features have on performance? In Eq. (12), the loss is calculated based on the difference between $z_0$ and $c$ , where multi-scale features do not appear to be necessary. Additionally, the authors have not demonstrated the role or necessity of multi-scale features in the ablation study. Clarifying these points would strengthen the paper’s argument for including multi-scale features.
>
> **R4**: We sincerely appreciate your constructive suggestions. To address your concerns, we will first present quantitative analyses through ablation experiments on the multi-scale feature structure:
>
> **Table 2.** The ablation experiments were conducted on the DIV2K dataset, utilizing the BD-rate (%) metric with DiffPC as the baseline.
> |      Model       | LPIPS | DISTS |  FID   |   KID    |   Paramters   |
> | :--------------: | :---: | :---: | :----: | :------: | :-----------: |
> | W/o Muti-feature | 5.455 | 7.286 | 6.9316 | 11.24091 | 24.53M (13%↓) |
> |  Ours (DiffPC)   |   0   |   0   |   0    |    0     |    28.20M     |
>
> It is evident that the removal of the Multi-feature component diminishes the model's performance. While this reduction may lead to a decrease in the model's parameter count, we deem it worthwhile to introduce a modest number of parameters to achieve a stable enhancement in performance. Furthermore, we have integrated the ablation experiments into Figure 8 in the main text and appended an analysis of this module.
>
> Subsequently, we present our perspective on this multi-scale structure.
>
> - Primarily, the utilization of multi-scale features, as demonstrated in various other works [1] [2], has been validated for its role in broadening the receptive field and preserving high-frequency information. Our intent is to augment the perceptual range of the encoding end towards the original image by amalgamating multi-scale features.
> - Furthermore, the original design intent of Stable diffusion for image generation directly results in the fabrication of high-frequency details in the generated images, rather than preserving them from the original image. These high-frequency signals are discarded during the encoding process by $\mathcal{E}(\cdot)$, whereas our designed multi-scale structure aims to fully recover these high-frequency details.
> - Your insightful observation regarding "the loss is calculated based on the difference between $z_0$ and $c$, where multi-scale features do not appear to be necessary" is astute. Indeed, in high-bitrate scenarios, multi-scale structures may not be essential as these additional high-frequency signals are likely to be discarded again due to the constraints of the optimization loss. **However, it is imperative to note that our rate-distortion loss is just a part of our training optimization objective, and these extra high-frequency signals are beneficial for optimizing the likelihood bound $\mathcal{L}_{CSD}$ of the denoising model**. Particularly in low-bitrate scenarios, where the distortion constraints are relatively weak, details originally contained in the signal $c$ are significantly discarded. **Through the multi-scale structure, we aim to complement these high-frequency details, prompting the compressor to focus more on textures rather than low-frequency semantics**. This is crucial as the low-frequency semantics are compensated for in subsequent stages through diffusion priors and semantic enhancements in the second phase.
>
> ---
>
> [1] Li, Feng, et al. "Lite detr: An interleaved multi-scale encoder for efficient detr." *Proceedings of the IEEE/CVF conference on computer vision and pattern recognition*. 2023.
>
> [2] Chang, Jia-Ren, and Yong-Sheng Chen. "Pyramid stereo matching network." *Proceedings of the IEEE conference on computer vision and pattern recognition*. 2018.
>
> ---
>
> **Q12**: In Eq. (13), how is the initial value of  $w$, specifically  $\sigma^2_{z_0}$, determined in practice? Additionally, how do you ensure that the trainable hyperparameter  $w$ correctly learns to achieve precise bit allocation?
>
> **R7**: Thank you for your comments. In fact, $\sigma^2_{z_0}$ is modeled during encoding in the analytical encoder $\mathcal{E}(\cdot)$ of Stable diffusion. $\mathcal{E}(\cdot)$ outputs both the mean and variance of the variable $z_0$, yet during training and sampling, typically only the mean is used as an estimate for the variable $z_0$. Hence, acquiring $\sigma^2_{z_0}$ is straightforward. Furthermore, we initialize the parameter $w$ with $\sigma^2_{z_0}$. During training, $w$ is updated; in the initial stages, the model focuses on rate-distortion optimization, resulting in minimal fluctuations in the value of $w$. As training progresses, slight updates are made to $w$ to align with the optimization objective of the diffusion loss $\mathcal{L}_{CSD}$. Overall, throughout the training process, $w$ consistently approximates the most accurate bit allocation, requiring no additional adjustments.

---

> > ### Author Response · Authors · 2024-11-21
> > **Author Response (Part III)**
> >
> > **Q13**:This module adopts a pre-embedding approach that combines textual semantics with visual semantics. Could the authors clarify the individual roles of textual semantics and visual semantics? Additionally, what impact would using only textual semantics or only visual semantics have on performance?
> >
> > **R6**: Thank you for your insightful commentary! We believe that there exists a disparity between high-level textual prompts and low-level image prompts during embedding, which explains why directly utilizing textual semantics hardly enhances the model's performance. We propose the pre-embedding structure for the following reasons:
> > - Introducing the pre-embedding module, we aim to narrow the gap between textual and image prompts in the embedding space through the pretrained q-former.
> > - Moreover, ICCN places greater emphasis on high-frequency components and disregards low-frequency semantics when receiving the image prompts $\hat{c}$. In the pre-embedding process, we seek to decouple and utilize the semantics contained in $\hat{c}$.
> > - Specifically, as the cross-attention interface of the pretrained denoising network is optimized based on text prompts, actively reducing the disparity between image semantics and textual semantics through the pre-embedding module also aids in faster convergence during training.
> >
> > To further address your inquiries, we have conducted quantitative ablation analyses:
> >
> > **Table 3.** The ablation experiments were conducted on the DIV2K dataset, utilizing the BD-rate (%) metric with DiffPC as the baseline.
> >
> >
> > |      Model       |  LPIPS  | DISTS  |   FID   |   KID   |
> > | :--------------: | :-----: | :----: | :-----: | :-----: |
> > | W/o img semantic | 19.8428 | 47.524 | 30.302  |  60.66  |
> > | W/o txt semantic | 9.7942  | 17.698 | 13.2731 | 34.3854 |
> > |      DiffPC      |    0    |   0    |    0    |    0    |
> >
> > It shows that both types of semantics contribute to performance improvements, with image semantics providing a relatively greater enhancement. In general, **image semantics offer more precise global information, encompassing color and style. Textual semantics provide coarser low-frequency information, contributing to the stability of the generated content. Another role of textual semantics is to assist in embedding image semantics more effectively within the denoising model.**

---

> ### Author Response · Authors · 2024-11-24
> **A Gentle Reminder of the Final Feedback**
>
> Dear Reviewer GPEP
>
> We deeply value your initial feedback and understand that you may have a busy schedule. However, we kindly request that you take a moment to review our responses to your concerns. Any feedback you can provide would be greatly appreciated. We are also available to address any additional questions before the rebuttal period concludes.
>
> Sincerely,
>
> The Authors

---

> > ### Comment · Reviewer_GPEP · 2024-11-26
> >
> > Thanks for your reply. My concerns are well addressed. I have raised the rating to 6. Good luck.

---

> ### Author Response · Authors · 2024-11-26
> **Thanks to reviewer GPEP**
>
> Thank you for your response! We are grateful for your recognition of the originality of our working methods and appreciate your efforts to enhance our work!

---

### Official Review · Reviewer_HiqS · 2024-10-31

**Soundness:** 2
**Presentation:** 3
**Contribution:** 3
**Rating:** 6
**Confidence:** 2

**Summary:**

This paper presents DiffPC, a two-stage image compression framework leveraging stable diffusion models. In the first stage, a multi-feature compressor encodes essential low-level details using minimal bitrates, while a pre-embedding module captures robust hybrid semantics by combining textual and visual information. In the second stage, DiffPC employs IC-ControlNet to ensure structural and textural consistency during image reconstruction, mitigating issues such as condition leakage. Extensive experiments demonstrate that DiffPC achieves state-of-the-art performance in both perceptual and statistical fidelity across various benchmark datasets, outperforming existing neural compression methods.

**Strengths:**

DiffPC introduces an framework by integrating diffusion models with semantic refinement for image compression  and the paper conducts thorough quantitative and qualitative evaluations across multiple datasets and compares DiffPC with several state-of-the-art methods .The results consistently show superior performance in both perceptual metrics and statistical fidelity, particularly at ultra-low bitrates.

**Weaknesses:**

Multi-step inference combined with heavy ControlNet and pre-trained base models pose significant computational challenges for the decoding side. While pre-trained T2I diffusion models can enhance the quality of compressed images, their high computational cost makes them an unacceptable solution for image compression.

**Questions:**

[1] Although latent diffusion can enhance semantic details, it can also compromise some structural details, such as small faces and small text, which are not demonstrated in the paper.

[2] I would like to understand the performance comparison between directly using VAE compression and DiffPC.

I will consider raising the score once all concerns are addressed.

---

> ### Author Response · Authors · 2024-11-21
> **Response to Reviewer HiqS**
>
> Thank you for your constructive feedback! We have observed that your concerns do not lie in the technical details of our approach but rather in the skepticism towards the use of pre-trained diffusion models in the technical roadmap. We would be more than willing to offer qualitative and quantitative experiments and explanations to help you better understand our framework.
>
> **W1**: While pre-trained T2I diffusion models can enhance the quality of compressed images, their high computational cost makes them an unacceptable solution for image compression.
>
> **R1**: We fully understand your concerns; indeed, multiple sampling does introduce significant decoding latency. We sincerely apologize for not analyzing the decoding complexity in the initial draft of the article. You can refer to the additional experiments included in our "Joint Response about Model Complexity" where we elucidate that the complexity of our approach is not "unacceptable."
> You can observe that our proposed solution maintains a comparable encoding latency on the encoding side and can reduce decoding latency by decreasing the sampling steps. Even with a sampling step of 50, we do not consider the decoding speed of DiffPC to be unacceptable. This is because the decoding latency of our approach is on par with Cheng20 and superior to the diffusion-based baseline Perco.
>
> In conclusion, **we deem our diffusion-based framework to be profoundly significant, particularly in addressing the challenge of cold data storage**. As elucidated in our manuscript, numerous enterprises house vast repositories of cold data—information that sees infrequent access but necessitates algorithms with high compression rates to mitigate storage expenditures. Put differently, in this scenario, prioritizing high compression rates and fidelity over real-time decoding proves to be more imperative.
>
> **Q1**：Although latent diffusion can enhance semantic details, it can also compromise some structural details, such as small faces and small text, which are not demonstrated in the paper.
>
> **R2**: Thank you very much for your inquiry. We understand your concerns and apologize for not addressing this issue adequately in the main text. In response to your query, we will provide qualitative experiments to elucidate. In the revised version's appendix, we present extra visual results regarding the "small faces" and "small text" you mentioned in Sec A.6.
>
> Next, we will address your question in the context of "low bit rates" mentioned in the paper.
> - Firstly, concerning the classification of "text", we categorize it into large, medium, and small classes. In the realm of small text, even the state-of-the-art neural compressors like ELIC, based on traditional rate-distortion optimization, fail to reconstruct legible text. Similarly, Gan-based methods [2,3] also fall short. With the premise of lost textual semantics during decoding, our proposed DiffPC decoding yields images with sharper edges and enhanced perceptual quality.
> - In the classification of 'medium text', all generative compression schemes exhibit some degree of text distortion. However, notably, GAN-based approaches [1,2] display more pronounced artifacts and structural deficiencies. In contrast, **DiffPC, while maximizing the preservation of textual semantics, maintains the details and textures of the remaining parts**. In the category of 'large text', all methods can successfully restore high-fidelity textual information.
>
> Similarly, we categorize faces into large, medium, and small classes, as depicted in the Fig.14.
> - In the large and medium categories, **DiffPC maintains remarkable structural consistency without distorted details**. Conversely, Gan-based approaches exhibit structural distortions, while ELIC shows severe blurring, making facial recognition challenging.
> - In the small face category, both Gan-based methods and ELIC demonstrate significant distortion and structural disarray. In contrast, DiffPC, at the expense of a certain level of semantic consistency, enhances overall structural coherence and texture details.
>
> Past research indicates a triad trade-off of distortion, realism, and bit rate in compression schemes under low bit-rate scenarios [3]. For the small text and small face scenarios you mentioned, existing compression approaches fall short of providing perfect reconstruction. However, we believe that our proposed DiffPC strikes the best balance between distortion and realism in these contexts.
>
> ---
>
> [1]Mentzer, Fabian, et al. "High-fidelity generative image compression." Advances in Neural Information Processing Systems 33 (2020): 11913-11924
>
> [2]Muckley, Matthew J., et al. "Improving statistical fidelity for neural image compression with implicit local likelihood models." International Conference on Machine Learning. PMLR, 2023.
>
> [3]Yochai Blau and Tomer Michaeli. Rethinking lossy compression: The rate-distortion-perception

---

> ### Author Response · Authors · 2024-11-21
> **Author Response (Part II)**
>
> **Q2**: I would like to understand the performance comparison between directly using VAE compression and DiffPC.
>
>
> **R3**: Thank you for your inquiry. We are uncertain if we have fully grasped your query; we present two interpretations:
> 1. What distinguishes the efficacy of our proposal from other VAE-based compression schemes?
> 2. How would the performance of DiffPC alter if the foundational denoising network were removed?
>
> - **Regarding the first interpretation**, the baseline model we compare, ELIC, is a neural compressor entirely structured upon VAE principles. Through experimentation, the significant performance gap between ELIC and DiffPC becomes readily apparent.
> - **For the second interpretation**, we have conducted ablation experiments to assist in clarifying any uncertainties you may have.
>
> **Table 1.** Remove noise network performance comparison. Tested on KodaK24 with a BD-rate (%) based on DiffPC (Ours)。
> |      Model       | LPIPS  | DISTS  |
> | :--------------: | :----: | :----: |
> | W/o denosing net | 871.92 | 960.41 |
> |       Ours       |   0    |   0    |
>
> It can be observed that the removal of the denoising network would have a catastrophic impact on overall performance, as the diffusion model prior we leverage stems directly from the denoising network. In low bit-rate scenarios, utilizing the prior from a pre-trained model can significantly enhance the quality of reconstruction [4]. **If we have misconstrued your inquiry, please inform us, and we will promptly address your concerns**. If you believe that additional experiments would be beneficial to address this issue, please let us know, and we will promptly provide further clarification for you.
>
> We trust that our explanations and experimental data can alleviate your apprehensions. We believe that compression schemes based on diffusion models hold great promise, whether at the image level or in compression tasks geared towards downstream applications, as diffusion models can offer robust priors as a powerful aid.
>
> ---
>
> [4] Careil, Marlene, et al. "Towards image compression with perfect realism at ultra-low bitrates." The Twelfth International Conference on Learning Representations. 2024

---

> > ### Comment · Reviewer_HiqS · 2024-11-27
> >
> > Thank you for the author’s response, which has addressed my primary concerns. However, I still have some questions regarding Author Response (Part II):
> >
> > 1. Why does retaining only the VAE for compression and reconstruction in Table 1 result in such poor outcomes? While it is understandable that encoding and decoding with a VAE may introduce some loss, is the degradation truly this significant?
> >
> > 2. Regarding the results in the second row, why are the LPIPS and DISTS values both reported as 0? This implies there is no any loss from encoding and decoding, which seems theoretically impossible. Even setting aside the impact of diffusion models on texture fidelity, the VAE alone should introduce some degree of compression loss.

---

> > > ### Author Response · Authors · 2024-11-27
> > > **Thank you for the response**
> > >
> > > We appreciate your invaluable feedback! We shall exert our utmost efforts to address your inquiries:
> > >
> > > ---
> > >
> > > **Q1**: Why does retaining only the VAE for compression and reconstruction in Table 1 result in such poor outcomes? While it is understandable that encoding and decoding with a VAE may introduce some loss, is the degradation truly this significant?
> > >
> > > **R1**:We believe that the significant performance decline in the pure VAE structure can be attributed to several key reasons:
> > >
> > > - As mentioned in 'Author Response (Part II),' the loss of the prior provided by the denoising network results in the decoded images lacking a substantial amount of texture and detail, containing only structural, color, edge information from the original images. **This outcome is expected, as it is challenging to encode complex texture information at rates ranging from 0.02bpp to 0.1bpp (240 compression ratio to 1200 compression ratio)**. The loss of this texture inevitably leads to a sharp decrease in perceptual metrics (lpips, dists. etc). *If you wish to see visualizations of the outputs without the denoising network, please inform us promptly, and we will add these visualizations before the manuscript revision deadline to alleviate your concerns.*
> > >
> > > - In our framework, the encoder $\mathcal{E}$ and decoder $\mathcal{D}$ are frozen, with their parameters optimized for diffusion model-based generation tasks. Therefore, purely using VAE encoding and decoding without additional loss functions jointly training $\mathcal{E}$ and $\mathcal{D}$ will also result in a performance decrease.
> > >
> > > **Certainly, if we were to train the encoder $\mathcal{E}$ and decoder $\mathcal{D}$ using Gan loss and perceptual loss, we would obtain a model similar to HiFiC [1]**. This is due to the fact that the encoder $\mathcal{E}$ and decoder $\mathcal{D}$ in Stable diffusion utilize the structure of VQ-GAN [2]. We regret not having adequate resources and time to fully train $\mathcal{E}$ and $\mathcal{D}$, **but you can observe in the experimental section that similar works like HiFiC [1] and its enhanced version MS-ILLM [3] exhibit noticeably lag behind DiffPC in terms of performance.**
> > >
> > > ---
> > >
> > > **Q2**: Regarding the results in the second row, why are the LPIPS and DISTS values both reported as 0? This implies there is no any loss from encoding and decoding, which seems theoretically impossible. Even setting aside the impact of diffusion models on texture fidelity, the VAE alone should introduce some degree of compression loss.
> > >
> > >
> > > **R2**:We apologize for the unclear statement, **as this issue is actually a misunderstanding**. As stated in the title of our Table 1, the metric we present is the **Bjontegaard Delta-Rate [4] (BD-rate), which represents the difference in bit rate between a method and a baseline method at the same distortion level**. In Table 1, we set our proposed DiffPC as the baseline, so its BD-rate is 0, **which does not imply no loss**. Conversely, solutions solely using VAE exhibit a positive BD-rate, indicating that significantly more bits are required for this method to achieve the same distortion (LPIPS, DISTS, etc) as DiffPC.
> > >
> > > ---
> > >
> > > Lastly, we thank you once again for your response. If you have any further questions, we would be delighted to continue the discussion with you.
> > >
> > > ---
> > >
> > > [1] Mentzer, Fabian, et al. "High-fidelity generative image compression." Advances in Neural Information Processing Systems 33 (2020)
> > >
> > > [2] Esser, Patrick, Robin Rombach, and Bjorn Ommer. "Taming transformers for high-resolution image synthesis." Proceedings of the IEEE/CVF conference on computer vision and pattern recognition. 2021
> > >
> > > [3] Muckley, Matthew J., et al. "Improving statistical fidelity for neural image compression with implicit local likelihood models." International Conference on Machine Learning. PMLR, 2023.
> > >
> > > [4] Gisle Bjontegaard. Calculation of average psnr differences between rd-curves. *ITU SG16 Doc.**VCEG-M33*, 2001.

---

> > > > ### Comment · Reviewer_HiqS · 2024-11-27
> > > >
> > > > Thank you for the clarification. I will increase the score and lean toward acceptance.

---

> > > > > ### Author Response · Authors · 2024-11-28
> > > > > **Thank You for Your Feedback**
> > > > >
> > > > > We appreciate your response. Thank you for engaging in discourse with us and for your recognition of our endeavors！

---

> ### Author Response · Authors · 2024-11-24
> **A Gentle Reminder of the Final Feedback**
>
> Dear Reviewer HiqS
>
> Thank you once again for dedicating your valuable time to reviewing our paper and providing constructive comments! As the end of the discussion period approaches, we kindly ask if our responses have satisfactorily addressed your concerns. If you have any further inquiries regarding our utilization of the diffusion model approach, or if you feel that your feedback has been misconstrued by us, please do not hesitate to inform us. We are more than willing to engage in timely discussions with you.Furthermore, if your questions have been addressed, we would be grateful if you could consider updating your rating accordingly.
>
> Sincerely,
>
> The Authors

---

> ### Author Response · Authors · 2024-11-27
> **A Second Reminder of the Post-rebuttal Feedback**
>
> We deeply value your initial feedback and understand that you may have a busy schedule. However, we kindly request that you take a moment to review our responses to your concerns.
>
> Utilizing the prior knowledge of diffusion models for image compression holds significant promise, and some works  [1] [2] [3] in this domain have garnered widespread recognition. We present additional visual results in the appendix, where you can clearly observe that our approach maintains exceptional fidelity even at ultra-low bit rates. Furthermore, the complexity of diffusion models is not sensitive to image resolution, which allows our method to demonstrate superior encoding and decoding speeds compared to sequence autoregressive entropy models when handling high-resolution images.
>
> In conclusion, we kindly request your response once more, as we are eager to engage in further discussion with you.
>
> ---
>
>
> [1] Towards Extreme Image Compression with Latent Feature Guidance and Diffusion Prior. IEEE TCSVT 2024.
>
> [2] Careil, Marlene, et al. "Towards image compression with perfect realism at ultra-low bitrates." The Twelfth International Conference on Learning Representations. 2024
>
> [3] Yang, Ruihan, and Stephan Mandt. "Lossy image compression with conditional diffusion models." *Advances in Neural Information Processing Systems* 36 (2024).

---

### Author Response · Authors · 2024-11-21
**Joint response  about model complexity**

We concur with the reviewer's suggestion to augment the experiments concerning the intricacy of encoding and decoding. We believe that supplementing this experiment would be advantageous to our manuscript.
- Apart from Perco, all our other baselines were tested using the same Nvidia 3080ti GPU. Due to the additional VRAM requirements of Perco, we conducted the tests using the Nvidia A6000 GPU, which possesses higher GFLOPS. Hence, in the table, we have utilized the symbol ">" to signify that the expected encoding and decoding time of Perco on the 3080ti GPU is greater than the values we have reported.
- For all the baselines, we calculate the runtime for all the Kodak images at a resolution of 512x768 and present the average time in seconds. We include these experiments in the revised Sec A.5

**Table 1.** Encoding and decoding delays on the gpu. Test latency on Kodak.

 BD-rate is calculated on CLIC2020 dataset, with FID as the metric.

|      Model      | Encoding (s) | Decoding(s) | BD-rate (%) $\downarrow$ |
| :-------------: | :----------: | :---------: | :--------: |
|     Cheng20     |    3.081     |    6.678    |  >10^(3)   |
|      HiFiC      |    0.013     |  3*10^(-5)  |   289.26   |
|     MS-ILLM     |    0.069     |    0.068    |   79.83    |
|  VQ-GAN based   |    0.011     |    0.011    |     -      |
|      TACO       |    0.121     |    0.144    |   62.96    |
|       CDC       |    0.007     |    3.080    |   143.54   |
|     DiffEIC     |    0.430     |    6.619    |   10.45    |
|      Perco      |    >0.767    |   >15.261   |     -      |
| Ours (steps=50) |    0.089     |    7.325    |     0      |
| Ours (steps=20) |    0.089     |    3.378    |    6.25    |
| Ours (steps=5)  |    0.089     |    0.886    |   14.59    |

- The term "steps" denotes the sampling steps required for decoding, with our default sampling step set to 50 in the paper. It is evident that by encoding only the latent code on the encoding side, DiffPC exhibits satisfactory encoding speed: surpassing most diffusion baselines and showcasing similar encoding latency to Gan-based methods. On the decoding side, we can actually reduce the sampling steps to decrease decoding latency. Even when we reduce the sampling steps to 5, DiffPC maintains superior performance and demonstrates considerable decoding speed. With a sampling step of 50, DiffEIC's decoding speed surpasses that of Perco, a diffusion-based model, and is comparable to the decoding speed of Cheng20, a neural compressor based on sequential autoregressive entropy models.

---

### Author Response · Authors · 2024-11-24
**Modifications to the revision**

We have revised the initial draft by incorporating the baseline as requested by the reviewers and by including more detailed ablation experiments between modules. Furthermore, we have rectified some figures and statements to enhance the clarity of the paper, thus averting misunderstandings. Specifically, our revisions are as follows:

**Experiment Revisions**:

- In Section 4.2, we introduced a new baseline[1] [2] [3],  along with additional experimental analyses. The introduction of this new baseline does not compromise the superiority of our proposed approach.
- In Section 4.3, we augmented the ablation experiments concerning text semantics and the multi-scale compressor module.
- In Appendix A.5, we incorporated experiments and analyses regarding encoding and decoding complexities.
- In Appendix A.6, we included visualizations and qualitative analyses for special images (including faces and text).
-  In Appendix A.7.1, we included ablation experiments on sampling steps, showcasing the decoding performance differences of DiffPC when using varying sampling steps.
- In Appendix A.7.2, we added an analysis on text robustness.
- In Appendix A.7.3, we included visual results on compressor bit rate allocation to further elucidate the contribution of multi-scale structures in optimizing bit rate allocation.

**Figure and Description Revisions**:


- We have revised typo errors and formatting mistakes in accordance with the reviewers' feedback.
- Figure 2 has been altered to include more detailed module annotations and symbol explanations.
- Figure 9 has been revised to incorporate the previously omitted residual module and complete the schematic diagram for this module.
- Figure 10 has been modified to rectify symbol descriptions and incorrect details.
- Errors in the spelling of the module TAD in Figure 8 have been corrected.
- The description of equation (3) has been adjusted to align more closely with a standard definition.
- Equation (7) has been modified to better reflect the description of the formula in accordance with our algorithmic process.

---

[1] Zhiyuan Li, Yanhui Zhou, Hao Wei, Chenyang Ge, and Jingwen Jiang. Towards extreme image compression with latent feature guidance and diffusion prior. IEEE Transactions on Circuits and Systems for Video Technology, 2024.

[2] Hagyeong Lee, Minkyu Kim, Jun-Hyuk Kim, Seungeon Kim, Dokwan Oh, and Jaeho Lee. Neural image compression with text-guided encoding for both pixel-level and perceptual fidelity. In International Conference on Machine
Learning, 2024

[3] Mao, Qi, et al. "Extreme image compression using fine-tuned vqgans." 2024 Data Compression Conference (DCC). IEEE, 2024.

---

### Meta-Review · Area_Chair_LCmz · 2024-12-18

**Metareview:**

This paper receives a mixed rating of (5, 6, 6, 6). The reviewers generally acknowledge the strong performance of the proposed DiffPC in the task of image compression, while the novelty of the proposed method receives mixed comments. After carefully reviewing the paper, review, and rebuttal, the AC agrees that the proposed method achieves decent performance, and the proposed modules have values to the research community. Therefore, the AC recommends an acceptance, with the suggestion that grammatical errors and minor writing flaws should be resolved in the camera ready version.

**Additional Comments On Reviewer Discussion:**

The reviewers raised questions in different aspects, including complexity, novelty, and technical details. The authors carefully address them with experimental results and explanations. After reading the responses, most of the AC's concerns have also been resolved, and the AC believes that this deserves an acceptance.

---

### Decision · Program_Chairs · 2025-01-22

Accept (Poster)